



# Assessing the potential for simplification in global climate model cloud microphysics

Ulrike Proske[1], Sylvaine Ferrachat[1], David Neubauer[1], Martin Staab[2], and Ulrike Lohmann[1]

[1]Institute for Atmospheric and Climate Science, ETH Zürich, Zürich, Switzerland
[2]Max-Planck-Institut für Gravitationsphysik (Albert-Einstein-Institut), Hannover, Germany

**Correspondence:** Ulrike Proske (ulrike.proske@env.ethz.ch)

**Abstract.** Cloud properties and their evolution influence Earth's radiative balance. The cloud microphysical (CMP) processes that shape these properties are therefore important to be represented in global climate models. Historically, parameterizations in these models have grown more detailed and complex. However, a simpler formulation of CMP processes may leave the model results mostly unchanged while enabling an easier interpretation of model results and helping to increase process under-

standing. This study employs sensitivity analysis on an emulated perturbed parameter ensemble of the global aerosol-climate model ECHAM-HAM to illuminate the impact of selected CMP cloud ice processes on model output. The response to the phasing of a process thereby serves as a proxy for the effect of a simplification. Aggregation of ice crystals is found to be the dominant CMP process in influencing key variables such as the ice water path or cloud radiative effects, while riming of cloud droplets on snow influences mostly the liquid phase. Accretion of ice and snow and self-collection of ice crystals have a

negligible influence on model output and are therefore identified as suitable candidates for future simplifications. In turn, the dominating role of aggregation suggests that this process has the greatest need to be represented correctly. A seasonal and spatially resolved analysis employing a spherical harmonics expansion of the data corroborates the results. This study introduces a new framework to evaluate a processes' impact in a complex numerical model, and paves the way for simplifications of CMP processes leading to more interpretable climate models.

## 1   Introduction

Aerosols and cloud microphysics (CMPs) control cloud properties and thereby exert a large influence on Earth's climate. For example, the cloud water and ice contents determine the cloud albedo and lifetime, and also control precipitation formation (Mülmenstädt et al., 2015). In a changing climate, the correct representation of clouds is especially important to estimate

Earth's radiation budget (Sun and Shine, 1995; Tan et al., 2016; Matus and L'Ecuyer, 2017; Lohmann and Neubauer, 2018). However, clouds and cloud feedbacks are a major source of uncertainty for projections of climate sensitivity in global climate models (Cess et al., 1990; Soden and Held, 2006; Williams and Tselioudis, 2007; Boucher et al., 2013).





Since cloud microphysical processes such as the riming of cloud droplets on snow flakes occur on scales much smaller than the resolution of global climate models (GCMs), they are parameterised, i.e. only their macroscopic effects at the scale of the
model grid are described. Responding to the challenge of incorporating these processes in climate models, the community has added more and more processes into GCMs (Knutti and Sedláček, 2013) with increasing detail in their representation (e.g. Archer-Nicholls et al. (2021)). As Fisher and Koven (2020) argue, this is due on the one hand to scientists' tendency to focus on their own area of expertise. On the other hand, it also reflects the fact that the Earth system is indeed complex and that many processes may matter. However, it is doubtful whether more detail will help us reduce uncertainty (Knutti and Sedláček,
2013; Carslaw et al., 2018). More complexity has also its downsides: More parameterised processes lead to more parametric uncertainty which in turn scientists investigate and try to reduce with large scientific effort (e.g. Rougier et al. (2009); Lee et al. (2011); Yan et al. (2015); Williamson et al. (2015); Dagon et al. (2020)). In fact, Reddington et al. (2017) argue that "aerosol-climate models are close to becoming an overdetermined system with many interacting sources of uncertainty but a limited range of observations to constrain them", refering to the complexity in the representation of aerosols and their interaction with
clouds. This is related to equifinality, meaning that model versions from different regions of the input parameter space may lead to the same results that compare well with observations. These models may simulate a range of aerosol forcings (Lee et al., 2016), which is not possible to constrain with current observations. Climate models have become so complex that they are impossible to comprehend by any one scientist (Fisher and Koven, 2020). More detail means more heterogeneity between climate models, which increases the difficulty of a meaningful comparison of their projections (Fisher and Koven, 2020).
But also within a given model, the attention and detail given to some cloud microphysical processes comes at the expense of other less accessible processes. This brings the danger of overinterpreting those processes that are represented in detail while neglecting the impacts of poorly represented ones (Mülmenstädt and Feingold, 2018). Finally, the detail of the aerosol and cloud microphysics increases computational demand and thereby costs or inhibits other advancements such as the move towards high-resolution simulations or larger ensembles (though anticipating the results of Sec. 3.6, the four CMP processes
investigated in this study require negligible computing time).

In contrast, simple models are easier to interpret and derive understanding from, as long as they represent processes correctly (Koren and Feingold, 2011; Mülmenstädt and Feingold, 2018). Also, assumptions and their consequences are more traceable in simpler or more system-oriented models (Mülmenstädt and Feingold, 2018). For example, conceptual cloud models have been used to investigate the impact of the choice of precipitation particles' attributes on the cloud structure and evolution (Wacker,
1995); or to confirm microphysics findings qualitatively (Wood et al., 2009). Simplifications reduce computational demand, and simplified models yield themselves to other applications, e.g. the use in integrated assessment models (Ghan et al., 2013). At the same time, they may produce similarly good results as more complex models. For example, Ghan et al. (2012) have developed a simple yet physical model for the aerosol indirect effect, whose estimates are comparable to that of complex global aerosol models. Similarly, Liu et al. (2012) compared two aerosol modules with seven and three lognormal modes and find that
the simulated aerosol concentrations are remarkably similar.

The addition of detail and refinement of a model description is a natural response to the challenge of capturing something as complex as the climate system in a computer model. This is legitimate and beneficial. For example, it may lead to a





physically more correct representation and reduce the amount of tuning parameters (e.g. Storelvmo et al. (2008)). And for some applications modelers may need as much detail as possible in one specific module. Hence, scientists tend to call for

more detail in process representations (e.g. Gettelman et al. (2013) for warm-rain microphysics; Sotiropoulou et al. (2021) for secondary ice production by break-up from collisions between ice crystals) instead of less. This may in part be because humans are biased towards searching for additive pathways as problem solutions while overlooking subtractive transformations (Adams, 2021). However, due to the reasons mentioned above, a simplified model equifinal to a more complex model may be more useful for gaining understanding of climate models. One can therefore question the need for an ever increasing amount

of detail, especially in face of overdetermination (Reddington et al., 2017). In this paper, we propose a new methodology to assess where process parameterisations can be less accurate, i.e. stripped of detail, to aid the development of a simplified model as well as to increase process understanding.

The role of CMPs within GCMs has been investigated previously: The influence of CMPs has been shown to dominate over that of aerosol schemes (White et al., 2017), as well as to dampen the influence of aerosol microphysics on cloud condensation

nuclei and ice nucleating particles (Glassmeier et al., 2017). Diving into the importance of single processes on the overall CMPs, Bacer et al. (2021) extracted process rates from the chemistry-climate model EMAC, which is based on the same CMPs as this study's ECHAM-HAM. They found that ice crystal sources in large-scale clouds are controlled by freezing and detrainment from convective clouds, while sinks are dominated by aggregation and accretion. This approach is somewhat similar to a pathway analysis (e.g. Schutgens and Stier (2014); Dietlicher et al. (2019)) in that it deepens understanding of

immediate effects, but is not able to relate the effect of a process on variables further down the process chain.

A promising method for investigating the effect of model input on output is the use of perturbed parameter ensembles (PPEs) (Murphy et al., 2004; Collins et al., 2011). In a PPE multiple input parameters are perturbed at the same time. In this way, PPEs are expanding upon sensitivity studies that vary one parameter (e.g. Lohmann and Ferrachat (2010)) or multiple parameters at a time (e.g. Ghan et al. (2013)), allowing to investigate the interaction effects of perturbations within the whole possible

parameter space. For example, Sengupta et al. (2021) used a PPE to determine the impact of parameters related to secondary aerosol formation on organic aerosol in a global aerosol microphysics model.

Another benefit is that a PPE does not require any additional changes to model code, in contrast to a pathway analysis that requires additional diagnostics and tracers. The downside is that PPEs require many simulations to sample the whole parameter space. A remedy is the combination of a PPE with a surrogate model such as an emulator. The emulator is first fitted to the

PPE model output and then sampled instead of the GCM which is expensive to run. This technique has been used for example to study the effect of model parameters such as the entrainment rate coefficient on climate sensitivity in a GCM (Rougier et al., 2009); or how model parameters affect forecast model drift (Mulholland et al., 2017).

Global sensitivity analysis is a method to quantify the effect of inputs on model output more formally. It allows to divide the total variation in output into the direct contributions from variations in input as well as to their interactions. For example,

Tan and Storelvmo (2016) used variance-based sensitivity analysis on a generalized model of their PPE to determine that the Wegener-Bergeron-Findeisen time scale is the most influential parameter in determining the cloud phase partitioning in mixed-





phase clouds. Bernus et al. (2021) have employed sensitivity analysis on their PPE directly to improve the understanding of their lake model prior to its implementation into a climate model.

When dealing with large models that are expensive to run, a surrogate model that is cheap to run allows for a tight sampling
of the whole parameter space which permits for sensitivity analysis on the resulting surface. As such, the combination of a PPE with a surrogate model upon which sensitivity analysis is performed has found wide use in cloud simulation studies (Wellmann et al., 2018; Glassmeier et al., 2019; Wellmann et al., 2020; Hawker et al., 2021a). For example, Lee et al. (2011) emulate a global aerosol model and find that the cloud condensation nuclei concentration in polluted environments is dominated by sulphur emissions, but that in remote regions interactions between different parameters are substantial. In particular, a range
of recent studies has employed the methodology to investigate how uncertainty in input parameters (which are often not well constrained within parameterisations) translates to an uncertainty of climate model output: quantifying the effect of aerosol parameters on cloud properties or radiative forcing (Lee et al., 2011, 2012; Carslaw et al., 2013; Lee et al., 2013; Regayre et al., 2014; Johnson et al., 2015; Regayre et al., 2015; Yan et al., 2015; Regayre et al., 2018), but also in various other areas of environmental modelling (e.g. a land model in Dagon et al. (2020)). In a further step, the effect of an observational
constraint on the model output can be investigated with the emulator/surrogate models (Tett et al., 2013; Williamson et al., 2013; Lee et al., 2016; McNeall et al., 2016; Johnson et al., 2018), yielding important conclusions about which observations are needed to constrain climate models and on which parameters we need to focus research efforts. The approach also lends itself to an investigation of tuning parameters since these also form a parameter space that needs to be explored and constrained (Williamson et al., 2015; Hourdin et al., 2020; Couvreux et al., 2021).

Here we propose a new application of the combined PPE and sensitivity analysis approach to learn about the needed accuracy in process parameterisations within GCMs. Instead of varying parameters within parameterisations, we phase in or out the processes themselves, as a whole. By phasing we mean that we vary the effectiveness of a given process, going from using 0 to 200% of a process's effect in the model. From the resulting response surface we infer the sensitivity of model output to the CMP processes. The thus generated understanding points to processes whose representation needs to be accurate since
they have a large influence, and suggests to simplify those processes that have little influence on model output. Accepting the notion of equifinality, we aim to identify those parts of our current model that do not contribute to variation in output. Thus, we develop a "global sensitivity analysis that can weed out unimportant parameters" as called for by Qian et al. (2016).

To avoid misunderstanding: we are using a surrogate model to learn about sensitivities within the ECHAM-HAM GCM. We are not aiming to replace CMP parameterizations with machine learned substitutes (as e.g. Seifert and Rasp (2020)) or
substitute model components (as e.g. Beusch et al. (2020)), because interpretable, physics-based models should be preferred (Rudin, 2019). Instead, in line with Couvreux et al. (2021) we are using emulation and sensitivity analysis as a tool to generate understanding that allows to build a more interpretable model version in a second step.

In the following Sect. 2 the CMP processes that we investigate, their treatment in the ECHAM-HAM GCM, the generation of the PPE and emulator as well as the sensitivity analysis are described. In Sect. 3 the results from a "one-at-a-time" sensitivity
study that explores the axes of the parameter space (Sect. 3.1), the emulated PPE (Sect. 3.2), and of the sensitivity study on the fully sampled parameter space (Sect. 3.3) are presented and discussed. Conclusions and an outlook are given in Sect. 4.





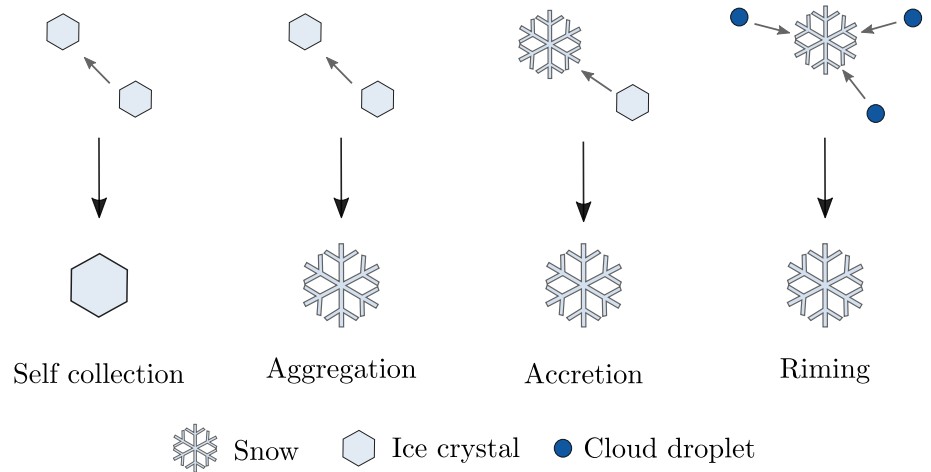

**Figure 1.** The four cloud microphysical processes investigated in this study, depicted as they are represented in ECHAM-HAM.

## 2   Methods

### 2.1   Cloud Microphysics in ECHAM-HAM

This study employs the global aerosol-climate model ECHAM6.3-HAM2.3 (Tegen et al., 2019; Neubauer et al., 2019), with a

T63 horizontal spectral resolution and 47 vertical levels. The cloud microphysics consist of a 2-moment prognostic scheme for ice crystals and cloud droplets, with additional 1-moment prognostic representation of snow and rain (Lohmann and Roeckner, 1996; Lohmann et al., 1999; Lohmann, 2002; Lohmann et al., 2007; Lohmann and Hoose, 2009; Lohmann and Neubauer, 2018). The stratiform cirrus scheme includes homogeneous nucleation of supercooled liquid droplets (Kärcher and Lohmann, 2002a, b; Lohmann, 2003). The stratiform liquid cloud scheme encompasses condensation, aerosol activation, autoconversion

of cloud droplets to rain as well as accretion of cloud droplets by rain, evaporation of cloud and rain water, and wet scavenging of aerosol particles (for further details and references see Neubauer et al. (2019). In stratiform mixed-phase clouds, various CMP processes are included: heterogeneous nucleation via immersion and contact freezing, depositional growth of cloud ice, growth of ice crystals at the expense of cloud droplets via the Wegener-Bergeron-Findeisen process (Wegener, 1911; Bergeron, 1935; Findeisen, 1938), sublimation and melting of ice crystals and snow below clouds. In this study, we are investigating

the effect of four different CMP processes involving the ice phase (see Fig. 1): **Self collection** of ice is the process of ice crystals sticking together to form a single ice crystal. **Aggregation** also has two ice crystals sticking together, albeit forming a snow flake. In **accretion**, a snow flake collects an ice crystal, resulting in a larger snow flake. The fourth process is the only one involving the liquid phase: Cloud droplets are **riming** on a snow flake, again enhancing its size. The implementation of these processes in terms of changes to the ice crystal and cloud droplet mass is detailed in Lohmann and Roeckner (1996),

while the implementation of changes to the ice crystal and cloud droplet number concentration is simply in proportion to the mass changes (Lohmann et al., 1999; Lohmann, 2002). The distinction between accretion and aggregation is necessary due



to the separation between ice crystals and snow flakes in their representation as categories of ice in the model. Snowflakes precipitate, while ice crystals are smaller and remain within clouds. The four processes were chosen for their comparability as they all represent particle interactions, to represent a range of assumed impacts, as well as for their implementation which is well distinguishable in the code and allowed for easy implementation of the phasing (see Sec. 2.2). In this study, we do not include any ice multiplication processes. Convective clouds are treated separately from stratiform clouds, except for the interaction through detrained condensate from convective clouds, which is added to stratiform clouds if they exist at the respective model level.

Apart from the phasing described in the next section, substantial changes that were applied with respect to the published model version ECHAM6.3-HAM2.3 (Neubauer et al., 2019) are:

- Detrained condensate from the convective cloud scheme produced an unrealistically large amount of ice crystals at mixed-phase temperatures, which were then removed with a correction term. The detrained cloud particles are now assumed to be liquid at mixed-phase temperatures (Dietlicher et al., 2019; Muench and Lohmann, 2020).

- In line with Muench and Lohmann (2020, Section 3.3.1.2), we now include the immediate, updraft dependent aggregation of detrained ice crystals.

- Previously, a fixed minimal cloud droplet number concentration (CDNC) was applied, which led to unrealistically high CDNCs in high latitude and/or high altitude clouds with low liquid water content (LWC) and hence small droplets. We replace this with a dynamically calculated minimal CDNC, which is calculated from the in-cloud water content and a set maximum volumetric cloud droplet radius (set at $15\,\mu m$ in the simulations conducted for this study). The resulting minimum CDNC needs to lie between $10^6\,m^{-3}$ and $4 \cdot 10^7\,m^{-3}$. Admittedly, we are replacing the tuning parameter of fixed minimum CDNC with one for a maximum cloud droplet radius. The latter is preferred as it is more physical.

- The model version of Neubauer et al. (2019) contains a mistake in the calculation of the hygroscopicity parameter in the aerosol activation parameterization, leading to an underestimation of the individual aerosol mode solubility. The calculation was updated in Friebel et al. (2019) and subsequently used in Lohmann et al. (2020) and this correction is also applied here.

- In part motivated by the large correction terms highlighted in the process rate study of Bacer et al. (2021) we reduce these if they are unnecessary and/or unphysical. For example, conditions of maximum ice crystal number concentration (ICNC) were enforced after a few CMP processes took place in Bacer et al. (2021). We could reduce the value of that correction term by applying it after each relevant process. Most importantly, the diagnosis of multiple correction terms acting on the same variable led to an artificial increase of corrections. For example, correction terms would enhance ICNC concentrations at model points that later were identified to be outside of a cloud (due to the way the code is structured, the diagnosis of cloud cover happens after e.g. activation/nucleation takes place). In turn, ICNC outside of a cloud were then corrected to be zero, so an unnecessary correction was in fact counted twice. We reduce this artifact by





correcting the correction terms themselves. Staying with the example above, the first correction term is now itself set to
zero outside of a cloud.

–  The sublimation of sedimenting ice crystals appears to be too weak in ECHAM-HAM. This became apparent as in-cloud
    ICNCs were increasing through sedimentation from above, which indicates that sublimation of ice crystals falling into
    the cloud-free part of a grid box is too weak. While the underlying problem of a weak sublimation needs to be addressed
    with future efforts, we introduced a correction of the sedimentation routine: the gain of ice crystal concentrations in
the level into which the ice crystals sediment is restricted to the closest loss of in-cloud ice crystal mass and number
    concentration in the levels above. Also, in-cloud ICNC concentration and snow formation rate are now set to 0 outside of
    clouds inside the ice crystal sedimentation routine where they were previously set to the grid-mean values. This contains
    the implicit assumption that ice crystals do not survive sedimentation outside of a cloud in ECHAM-HAM.

With the described changes, the model requires retuning. The tuning procedure follows the one described in Neubauer et al.
(2019), with the final tuning parameters given in Table A1 in the Appendix. Model simulations were conducted with the same
tuning for all simulations.

## 2.2  Phasing as a proxy for complexity

In order to see the effect of whole processes on model output, we can turn processes off in sensitivity studies. In the present
study, we achieve this by setting to zero the change that the process inflicts on tracer variables. For example, at every model
timestep $t$ aggregation impacts the ICNC:

$$\text{ICNC}_{t+1} = \text{ICNC}_t + \Delta\text{ICNC}_{\text{aggr}} \tag{1}$$

We can turn off the effect of aggregation by multiplying $\Delta\text{ICNC}_{\text{aggr}}$, the change in ICNC due to aggregation in one timestep,
by zero when it is added to the affected variables.

More generally, instead of setting to zero the changes inflicted by a process, we can phase these changes in and out using a
newly defined parameter $\eta$.

$$\text{ICNC}_{t+1} = \text{ICNC}_t + \eta_{\text{aggr}} \cdot \Delta\text{ICNC}_{\text{aggr}} \tag{2}$$

From the response of model output to variations in $\eta_i$, we can extract information on how accurately a process $i$ needs to be
represented in the model. For example, if the model output variable (e.g. ice water path, IWP) as a function of $\eta_i$ follows a
sigmoidal function, this suggests that the process $i$ needs to be represented only roughly and that some detail could probably be
removed from its parameterisation without much of an effect on the model performance. In this study, four cloud microphysical
processes, namely self collection, aggregation, accretion and riming (see Fig. 1) are phased, i.e. $i \in [1, 4]$. Combining pertur-
bations of multiple processes allows us to study and take into account possible interaction effects, such as the compensation by
one process which is perturbed by another one.





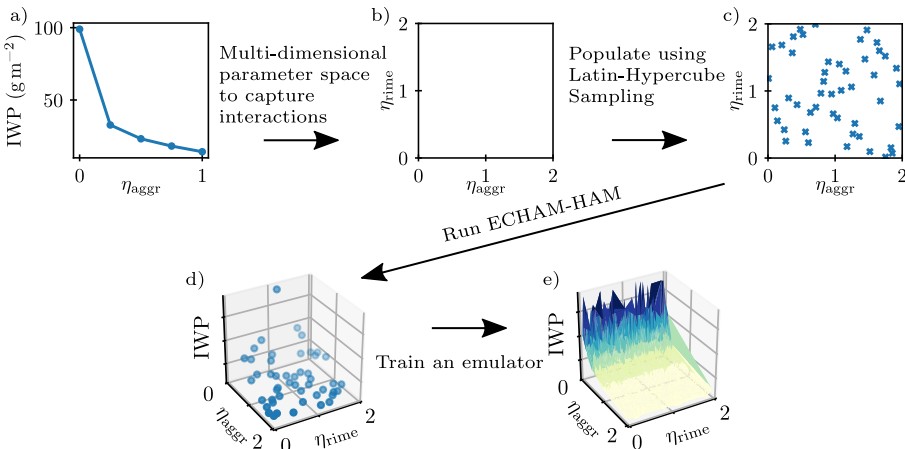

**Figure 2.** Sketch of the employed methodology: we move from **a)** one-dimensional sensitivity studies where one process is phased by varying the parameter $\eta$ (Sect. 3.1) to **b)** a multi-dimensional parameter space. **c)** The input parameter space is filled with Latin Hypercube sampling and supplied as input to ECHAM-HAM. The simulations form the perturbed parameter ensemble (PPE). The **d)** PPE output is **e)** fitted using a Gaussian Process emulator for each variable of interest to generate a smooth response surface, upon which sensitivity analysis can be applied. Note that this is an illustrative sketch of the method for a PPE with two input dimensions, whereas our PPE has four dimensions, and that the data used to generate it is only illustrative as well. The shading in **d)** illustrates depth only.

## 2.3 Generating and emulating the perturbed parameter ensemble (PPE)

In a first scoping study, we phase each process one by one, by multiplying its effect with $0 < \eta_i < 1$. Phasing between zero and one corresponds to a reduction in the process' effectiveness. However, to take into account interactions, all $\eta_i$ need to be varied at the same time, thereby creating a multi-dimensional input parameter space in a second step. In addition, $\eta_i$ is expanded to values up to $\eta_i = 2$ to imitate an overestimation of a given process due to an inacurate description. This and the procedure described in the following is visualized in Fig. 2. To probe the multi-dimensional input parameter space effectively, the sets

of simulation input were generated with Latin Hypercube Sampling (LHS, using the Python library PyDOE (tisimst, 2021)), which maximizes the spacing between inputs and provides good coverage of the parameter space, even when only a few input parameters are important (Morris and Mitchell, 1995). Each of the thus generated input combinations was then used as input for a 1 year ECHAM-HAM model simulation, creating a perturbed parameter ensemble (PPE) with 48 members. This is in line with the suggestion of Loeppky et al. (2009) to use 10 times as many training runs as the number of input parameters

for such a computer experiment. To estimate the inter-annual variability (IAV), the control simulation with all processes at full effectiveness ($\eta_i = 1 \forall i$) spanned 10 years. This estimate is used to judge whether perturbations observed in the PPE are significantly larger than the IAV and therefore contain a signal that originates from the perturbation in $\eta_i$. As the inter-annual variability exhibited no strong variations throughout the probed phase space in the one-at-a-time sensitivity studies, the 1 year simulations for the PPE members in combination with the control simulation estimate of the variability were deemed sufficient

for the analysis. All the simulations were performed with climatological sea surface temperatures and sea ice extents, and





aerosol emissions representative for the year 2003. These simulations were not nudged to meteorological data but ran freely so that the full effect of phasing the processes could be observed. Each simulation included a 3 months spin-up that was not included in the analysis.

Using the PPE output as input for the creation of a surrogate model, we can construct a smooth response surface over the whole parameter space (see Fig. 1e)). As a surrogate model, we choose a Gaussian process emulator (O'Hagan, 2006; Rasmussen and Williams, 2006), which has found wide use in atmospheric and climate science (Lee et al., 2011; Carslaw et al., 2013; Johnson et al., 2015). Using a recent Python package for emulating Earth System Models (Watson-Parris, 2021; Watson-Parris et al., 2021), the implementation is straight forward. From the PPE, we can construct a surrogate model for every output variable that we are interested in by training a separate emulator for each output variable (ice crystal and cloud droplet number concentration, ice and liquid water path, shortwave and longwave cloud radiative effect, cloud cover, precipitation, ice, liquid and mixed-phase cloud cover). As kernel, an additive combination of the linear, polynomial, bias and exponential kernel was used (Duvenaud, 2014). Other model specifics were set as default in Watson-Parris (2021). The input data was centered and whitened prior to emulation. With the cheap surrogate model a variance-based sensitivity analysis (see Sec. 2.5) becomes feasible (Oakley and O'Hagan, 2004), picking 3000 samples from the emulator as input. This approach is similar to Johnson et al. (2015), except that they perturb CMP parameters while we vary the effectiveness of whole CMP processes. It allows us to identify the importance of the different $\eta_i$ for the variables in question and thereby the processes which require a detailed representation.

## 2.4 Validation

To make sure that the chosen emulators are a fair representation of the model output, we validate them according to Bastos and O'Hagan (2009) except for using 1-out-validation, as visualized in Fig. 3 for the IWP. In Fig. 3 a) and b), the individual standardized errors, $\frac{Y_{\mathrm{sim}} - Y_{\mathrm{emu}}}{\sqrt{V_{\mathrm{emu}}}}$ (with Y the output of the ECHAM-HAM simulations and the emulated output, respectively, and $V$ the emulator variance), are plotted against the emulated output and input parameters. Errors larger than 2 signal a conflict. We observe only few such errors, which appear at high IWPs and $\eta_{\mathrm{aggr}}$ close to zero. This systematic conflict can be attributed to the threshold behaviour we investigate in Sec. 3.1. As such it is difficult to fit for the Gaussian Process emulator. As the special behaviour when turning off aggregation is explained in Sec. 3.1 and the emulator does not validate well in this part of the parameter space, we exclude it from the sensitivity analysis in Sec. 3.3. We employ a QQ-plot to determine whether the normality assumption of a Gaussian process is met in the emulator (Bastos and O'Hagan, 2009). The plot compares the quantiles of the standardized errors against those of a Student-t distribution. Fig. 3 c) indicates that the normality assumption holds and that the predictive variability is well estimated by the emulator (Bastos and O'Hagan, 2009), but the aforementioned outlier for large IWP is present. In a direct comparison of emulated and simulated ECHAM-HAM model output (Fig. 3d)), the points should lie close to the line of equality, with the 95% confidence bounds of the emulator crossing it. This should be the case for 95% of the validation points. In our emulations, the number of points with confidence bounds that do not cross the line of equality sometimes is larger (up to 33%), depending on the variable. We attribute this to the disruptive changes that the CMP process phasing induces as compared e.g. to the aerosol and CMP parameter changes applied by Johnson et al. (2015)



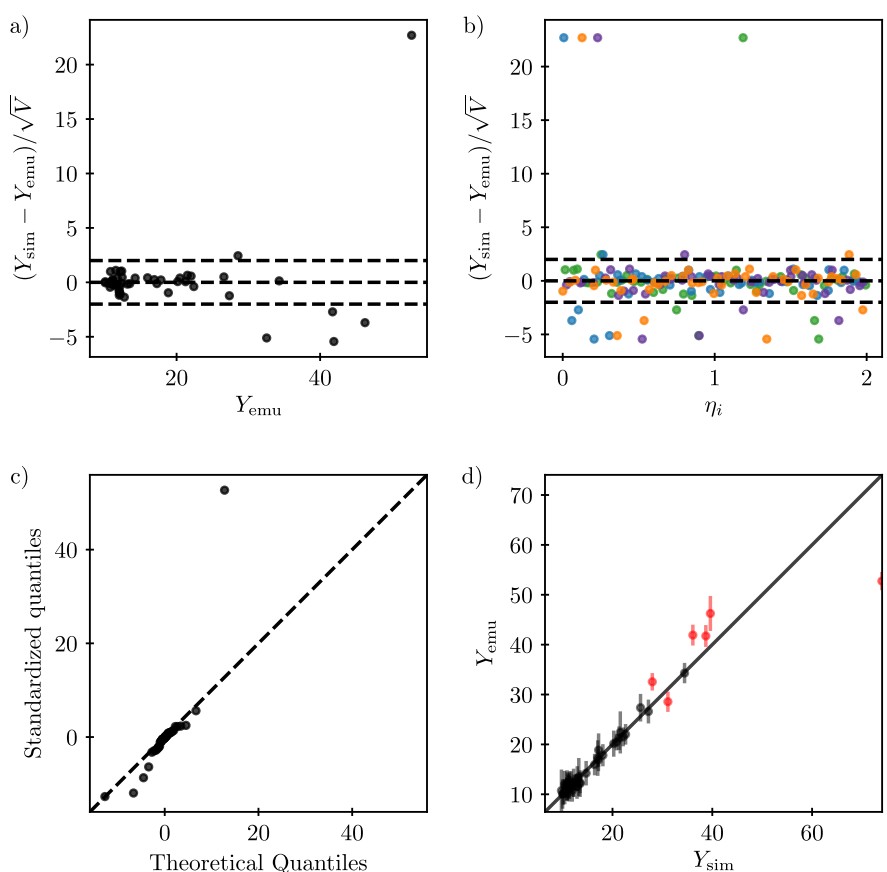

**Figure 3.** 1-out-validation of the emulator for global annual mean IWP according to Bastos and O'Hagan (2009). Each point corresponds to the training of the emulator on all points except one and then testing on exactly that point. Individual standardized errors are plotted against **a)** emulator output and **b)** input parameters (colors according to Fig. 4: aggregation (blue), accretion (purple), riming (green), self-collection (orange)). The dashed lines are drawn at an individual standardized error of zero and 2, which is the threshold discussed in Bastos and O'Hagan (2009). **c)** QQ-plot of the individual standardized errors against a student-T distribution. **d)** Emulator against model output, with the error bars indicating the 95% confidence interval on the emulator predictions. Predictions for which the model result lies outside that interval are marked red.





(which did not include ice cyrstal aggregation), as well as to the fact that the simulations were not nudged. Again we observe the threshold behavior investigated in Sec. 3.1, which is inherently difficult to capture with a Gaussian process emulator, provoking the striking outlier present in Fig. 3 for small $\eta_{\mathrm{aggr}}$. The difficulty in emulating the response surface for some of the variables was also apparent in computational limitations: some of the 1-out-validation emulations were not possible to compute because the constrain of the emulator was too tight for the variability in the data. As these were only few cases (up to one for global

means and seven for seasonal means in 48 validation emulations), the validation for those variables as a whole is still deemed valid.

Excluding the input space around $\eta_{\mathrm{aggr}} = 0$, the good qualitative agreement with the line of equality and the lack of systematic errors are sufficient for a validation of the emulator, especially considering that we are not aiming for exact quantitative estimates as results of the presented analysis. Rather, we are looking for a conceptual understanding of the need for an accurate

description of CMP processes, for which the thus validated emulators are sufficient.

For the variables which passed the 1-out-validation, the final emulator used for the sensitivity analysis was trained on all PPE members. Note that the setup of the emulator includes design choices such as the kernel combination to use. Therefore, the present emulator is only one of multiple possible emulators that could be used to represent the model data. However, as it is shown to validate well, other setups are expected to lead to the same conclusions as this one in the analysis.

## 2.5   Sensitivity analysis

In our framework, the question of how detailed the representation of a given process $i$ needs to be translates to the question of how sensitive the model output is to a variation of the phasing parameter $\eta_i$. For an answer, we employ variance based sensitivity analysis, following Saltelli (2008). In contrast to derivative-based local methods (Errico, 1997), global variance based sensitivity analysis allows for an investigation of sensitivities within the whole input parameter space. Its main metrics

are the first and total order sensitivity indices ($S_i$ and $S_{Ti}$, respectively). The first-order sensitivity index of $\eta_i$ measures the contribution of variance in $\eta_i$ to the variance in an output variable $Y$. It is constructed as

$$S_i = \frac{V_{\eta_i}\left(E_{\eta_{\sim i}}(Y|\eta_i)\right)}{V(Y)} \tag{3}$$

$E$ is the average over $Y$ with all $\eta$ except $\eta_i$ ($\eta_{\sim i}$) being allowed to vary while $\eta_i$ is kept fixed at $\eta_i^*$. Then $V_{\eta_i}$ is the variance over that average, for varying $\eta_i^*$. $S_i$ is always between 0 and 1, and high values signal an important variable. For additive

models all first-order terms add up to one, i.e. $\sum_i S_{\eta_i} = 1$. In non-additive models (e.g. a climate model) interaction terms also have to be taken into account. However, in models with many input parameters the computation of all interaction sensitivities can be cumbersome. The total effect sensitivity index $S_{Ti}$ offers a remedy in that it summarizes all direct and interactive effects a parameter's variance has on the total variance in output (Homma and Saltelli, 1996; Saltelli, 2008). It is defined as

$$S_{Ti} = \frac{V_{\eta_{\sim i}}\left(E_{\eta_i}(Y|\eta_{\sim i})\right)}{V(Y)} \tag{4}$$

Here all-but-$\eta_i$ ($\eta_{\sim i}$) are kept fixed at $\eta_{\sim i}^*$ and only $\eta_i$ is allowed to vary for the average $E_{\eta_i}$. Then the variance of that average over varying $\eta_{\sim i}^*$ is computed and divided by the variance in output $Y$. Saltelli et al. (1999) argue that the first and total

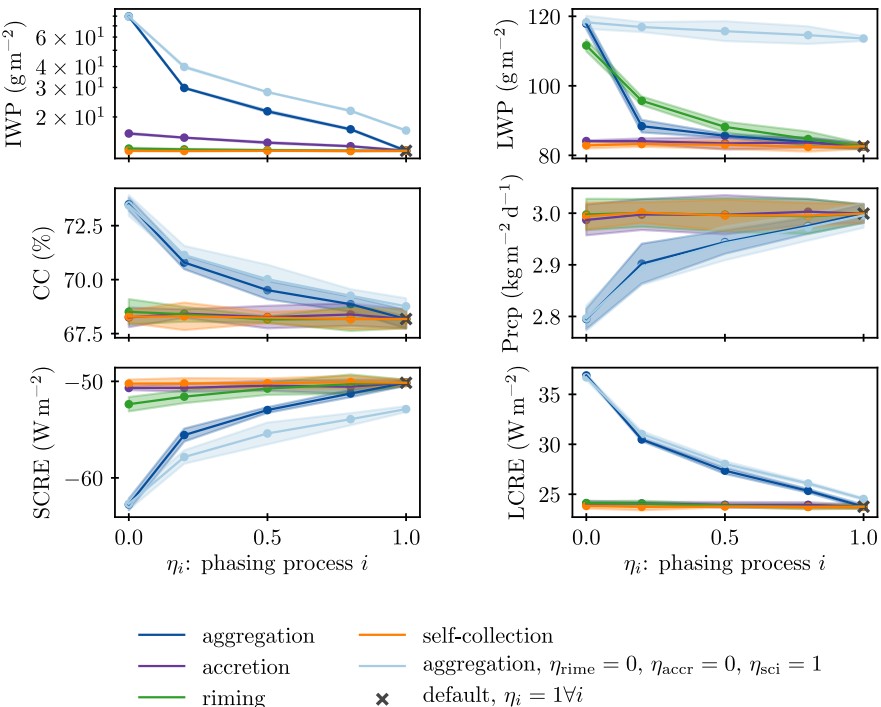

**Figure 4.** Model response to phasing four CMP processes: aggregation, accretion, riming and self-collection (as illustrated in Fig. 1) in terms of global annual mean IWP, liquid water path (LWP), cloud cover (CC), precipitation (Prcp), shortwave and longwave cloud radiative effect (SCRE, LCRE). An additional experiment was conducted to highlight interactive effects between the phasing of aggregation and the inhibition of riming and accretion (light blue). The points and line indicate the mean and the shading indicates 2 times the standard deviation of annual mean values of a five-year simulation. Classical sensitivity studies would only show $\eta_i = 0$ and $\eta_i = 1$. Note that for the IWP the shading is hidden behind the lines.

sensitivity index suffice for a meaningful global sensitivity analysis. To compute these indices via the Sobol method, we make use of the Python library SALib (Herman and Usher, 2017).

## 3 Results and Discussion

### 3.1 One-at-a-time sensitivity studies

In a first scoping experiment, we phased each process separately, which one can imagine as tracing the edges of the cube shown in Fig. 2. The results are presented in Fig. 4. Of the four phased processes, turning off aggregation has the largest effect on model output: the global annual mean ice water path (IWP) is more than doubled, and the increase in cloud cover and decrease in precipitation dwarf the changes inflicted by the inhibition of the other three processes. In fact, the perturbations inflicted by





phasing accretion and self-collection are mostly insignificant compared to the IAV. As aggregation is a removal process for ice crystals, it is reasonable that its inhibition leads to an increase of ice in the atmosphere (note that the IWP in ECHAM-HAM only counts ice crystals and not snow). Similarly, riming is a removal process for liquid droplets, so the liquid water path (LWP) increases with its inhibition. However, surprisingly the inhibition of aggregation inflicts a similarly large increase in LWP as that of riming, even though aggregation includes no direct interaction with liquid droplets. The shape of the model

response to the gradual phasing of the processes holds additional information: while the generated model response is mostly gradual, for low $\eta_{\mathrm{aggr}}$ the response is more abrupt. This is most striking for the global annual mean LWP, for which the signal for $\eta_{\mathrm{aggr}} \geq 0.25$ is not significantly different to that of accretion and self-collection. When aggregation is completely inhibited, the LWP increases dramatically and the signal becomes stronger than that for riming, which had increased consistently and gradually. This behaviour can be explained by aggregation acting as a threshold process for the CMPs. As can be seen from

Fig. 1 it is the only process that generates snow flakes. Accretion and riming need the snow flakes to be able to act upon them. Therefore, when aggregation is turned off, accretion and riming are consequently inhibited as well. In this way, the inhibition of aggregation can strongly influence even the liquid phase. The simulations in which we phase aggregation while having riming and accretion turned off confirm this hypothesis (light blue line in Fig. 4): Throughout most of the phase space, turning off accretion and riming reinforces the signal from phasing out aggregation. However, when aggregation is turned off, turning off

accretion or riming does not change the model output any further. That is because they are both inhibited when aggregation is turned off and does not generate any snow for them to act upon.

Fig. 5 further elucidates the reaction of the model to an inhibition of aggregation further: the snow formation rate decreases dramatically, and with increased ice concentrations in the atmosphere, the other removal processes of sedimentation and melting subsequently increase. Again the inhibition of riming and accretion only influences the model output when aggregation is

active. When aggregation is turned off, accretion and riming have no influence.

From this one-at-a-time example, one can already see the benefit of the phasing approach: In classical sensitivity studies, where processes are only turned on and off, only the large signal induced by aggregation would have been visible. However, here it was the peculiar shape of the model response to the whole phasing that hinted at the threshold effect of aggregation. The implications for possible simplifications are different: seeing only the large difference between a simulation with and without

aggregation, one would think that this is an immensely important process. Recognizing it as a threshold process and seeing the gradual response to small deviations from 1.0 in $\eta_{\mathrm{aggr}}$, it appears that there is potential for a less accurate description of aggregation in the model. It has also become clear that interaction effects need to be taken into account as well to explain the model behaviour. This is what the PPE expands upon in the next section.

## 3.2 PPE of global mean variables

Conducting a one year simulation with ECHAM-HAM for each of the 48 input parameter combinations generates the PPE which is then emulated (see Fig. 2). Fig. 6 illustrates the resulting response surface with points sampled from that emulation of the annual global mean IWP. To generate the multi-dimensional response surface 48 one year simulations were needed, compared to the 21 simulations that were needed to investigate the response along only a few of the parameter space edges in



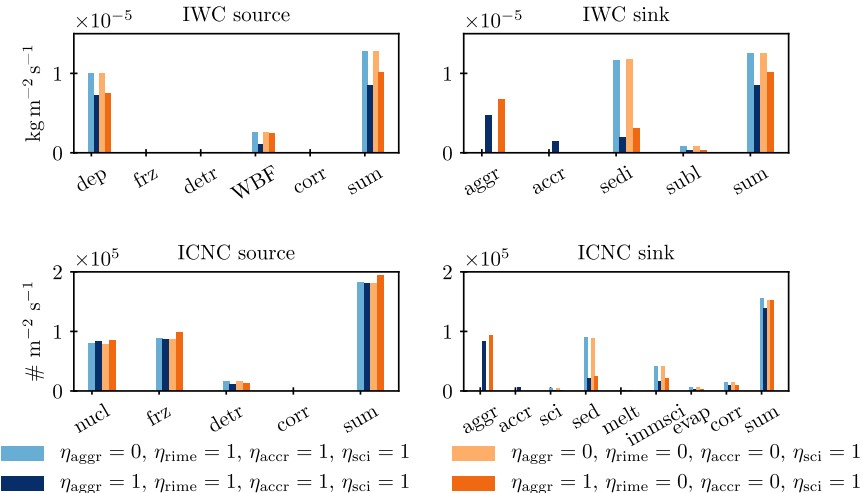

**Figure 5.** Global annual mean process rates for four experiments that illustrate the inhibition of snow formation through turning off aggregation (mean of a five-year simulation). The rates are diagnosed similar to Bacer et al. (2021), but correction terms were themselves subtracted from process rates where appropriate, i.e. where the correction belongs to the logical entity of the process rate (see Sec. 2.1). The process rates are: deposition (dep), heterogeneous and homogeneous freezing (frz), detrainment (detr), deposition in the Wegener-Bergeron-Findeisen process (WBF), correction terms (corr), aggregation (aggr), accretion(accr), sedimentation (sedi), sublimation (subl), ice nucleation in the cirrus scheme (nucl), melting (melt), immediate self-collection of ice crystals when the ICNC is larger than a maximal threshold (immsci), evaporation (evap).

Sec. 3.1. This illustrates the value of the chosen approach: the emulated PPE provides more information while needing only
roughly twice as many simulations. The surface shows an ordered ascent with decreasing $\eta_{\mathrm{aggr}}$, while the other dimensions exert no control over the value of the IWP. Only for accretion a slight influence is visible from the tilted contours in the phase space shared with aggregation. Increased accretion depletes the IWP since it converts ice crystals to snow flakes. Fig. 7 shows that the LWP is dominated by $\eta_{\mathrm{rime}}$, with an additional influence of aggregation. The LWP decreases with increasing $\eta_{\mathrm{rime}}$ and increasing $\eta_{\mathrm{aggr}}$. This is because riming depletes the atmosphere of cloud droplets and a decrease in aggregation inhibits
riming. The panels in Fig. 6 and 7 exhibiting no order in their parameter space distribution indicate that the processes in question exert no influence on the respective output variable. Similar to the LWP, the CDNC is dominated by riming, and for other cloud variables the dominant influence of aggregation is confirmed as well (see Fig. B1 in Appendix B).

The ranges in the global annual mean model variables that we observe are mostly larger than what Lohmann and Ferrachat (2010) find for varying uncertain tuning parameters, indicating that whole processes exhibit a larger influence on the model
response than those single parameters. Only for LWP Lohmann and Ferrachat (2010) find a larger range of about $50\,\mathrm{g\,m^{-2}}$ when they vary the autoconversion rate between 1 and 10. As this warm-rain process is not included in the present analysis, it is reasonable that the observed variation for LWP is smaller.





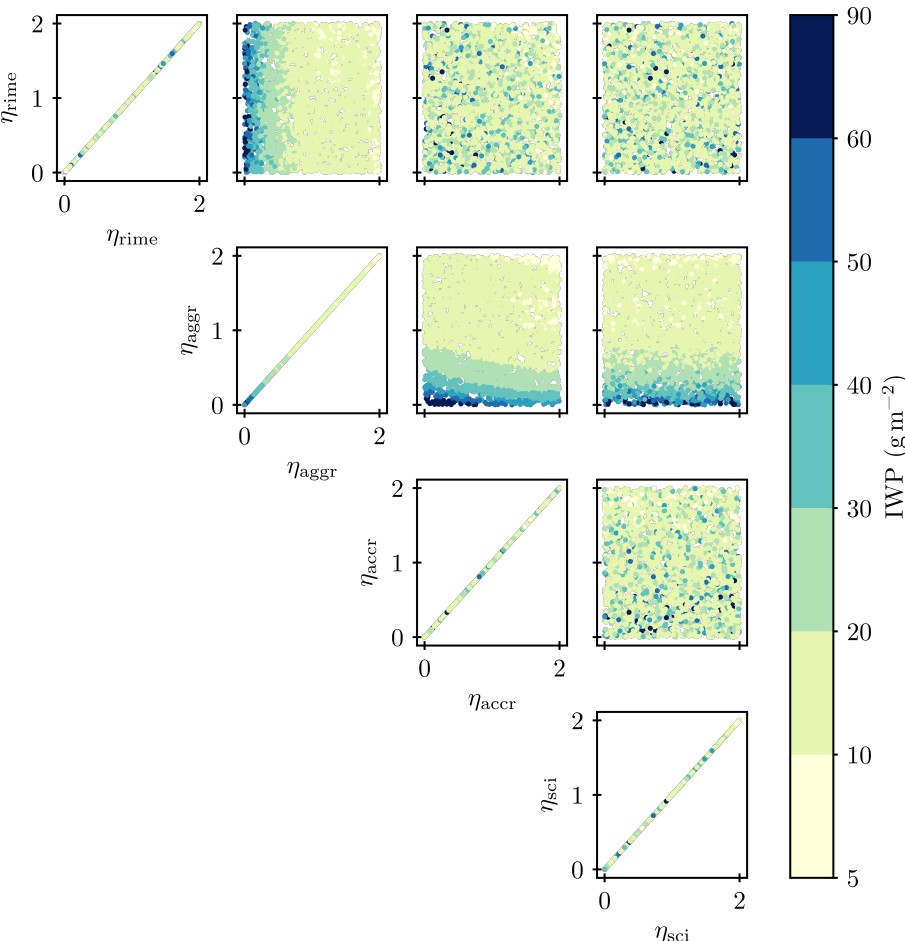

**Figure 6.** Correlation matrix as a visualisation of the multi-dimensional response surface of the emulated PPE. Each phased process is a dimension, and the colorbar denotes the global annual mean ice water path for each input parameter combination.

### 3.3 Sensitivity analysis

A global variance based sensitivity analysis allows to quantify the qualitative sensitivities obtained from the graphical repre-
sentations of the emulated surfaces in the previous section. The results for the first order ($S_i$) and total effect ($S_T$) sensitivity indices are presented in Fig. 8. Indeed, the qualitative results are confirmed: the global annual mean LWP and CDNC are dominated by riming while all other variables are dominated by aggregation, both in first order and total effect.

The observed sensitivities are different from what Bacer et al. (2021) find in their investigation of EMAC ICNC process rates. They find that aggregation contributes about twice as much as accretion to the ICNCs, while self-collection has a negligible
role. In our analysis, the influence of aggregation dwarfs that of accretion in terms of sensitivity indices as well as for the




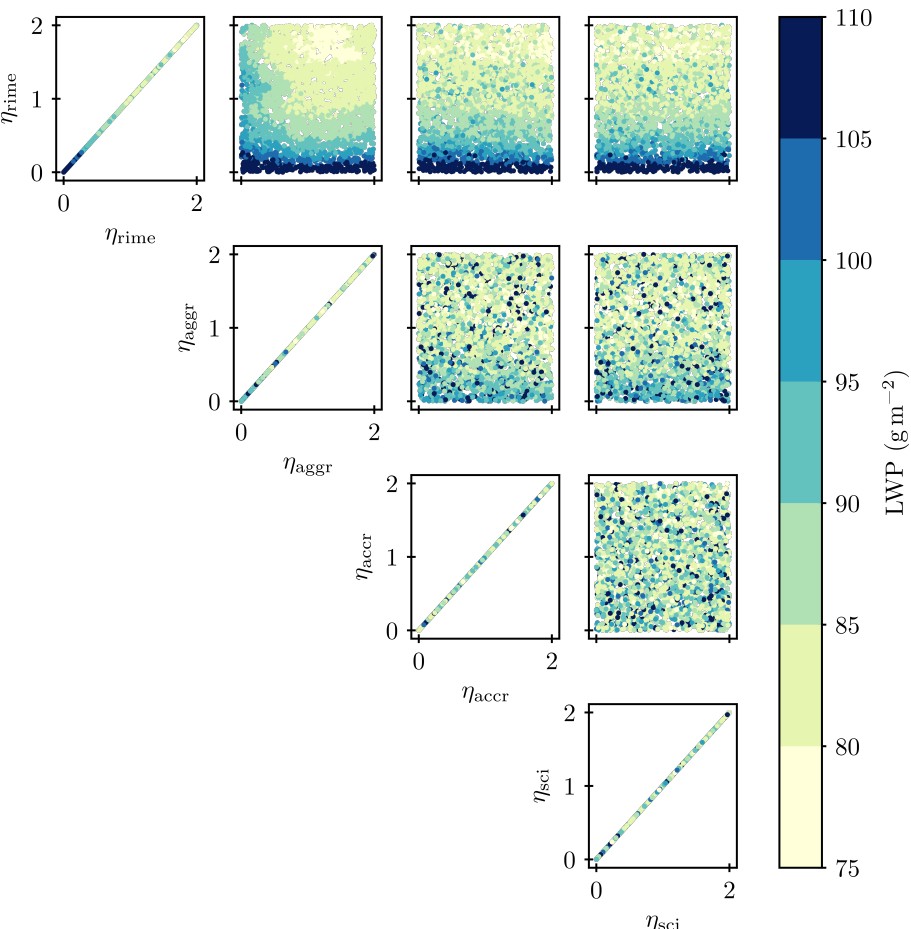

**Figure 7.** Same as Fig. 6 but for the global annual mean liquid water path. Correlation panels for additional variables are presented in Fig. B1 in Appendix B.

process rates (see Fig. 5). We attribute these differences to the slightly different model verion used in Bacer et al. (2021), which goes along with a different tuning.

The almost binary results for the sensitivity indices are surprising, as in other studies the sensitivity indices were more evenly distributed (Lee et al., 2011; Wellmann et al., 2018, 2020). However, these studies usually employed a wider suite of

input parameters, whereas here only processes from the limited system of ice particle interactions are included. We expect that with additional cloud microphysical processes included, the sensitivities would be more evenly distributed as well. The binary signal is due to the strong dominance of aggregation throughout the parameter space and not due the threshold behaviour upon inhibition of aggregation as analysed in Sec. 3.1. This was excluded from the sensitivity analysis as only the input parameter space with $\eta_{\mathrm{aggr}} \geq 0.5$ was taken into consideration (using the original emulation). In addition, note that the results including

the whole parameter space (see Fig. C1 in Appendix B) are similar.





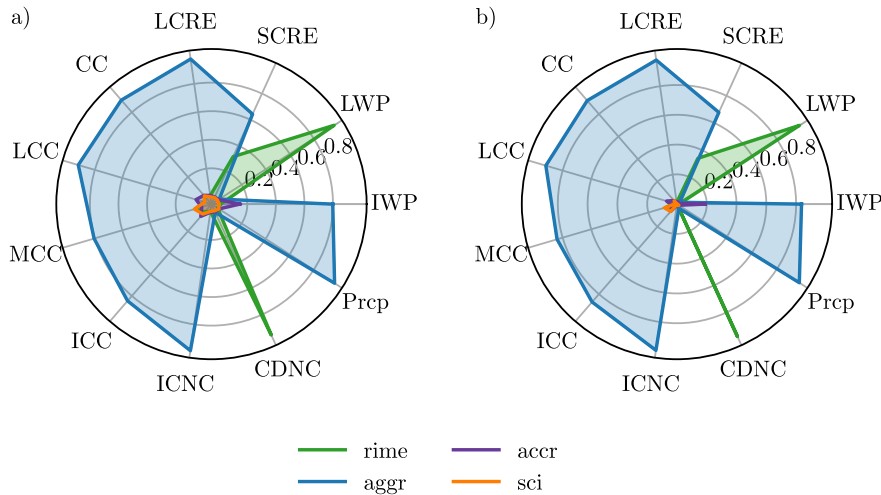

**Figure 8.** First order (**a**)) and total effect (**b**)) sensitivity indices for the emulated response surface of global annual mean cloud cover (CC), liquid cloud cover (LCC, $T > 0\,^\circ\mathrm{C}$), mixed-phase cloud cover (MCC, $0\,^\circ\mathrm{C} < T < -35\,^\circ\mathrm{C}$) ice cloud cover (ICC, $T < -35\,^\circ\mathrm{C}$), longwave cloud radiative effect (LCRE), shortwave cloud radiative effect (SCRE), liquid water path (LWP), ice water path (IWP), and total precipitation. As described in Sec. 2.5, the indices are always between 0 and 1, and high values signal an important variable. Since the climate model is non-additive, the terms do not add up to one as interactions have to be taken into account. The sensitivity analysis was applied only to the response surface with $\eta_{\mathrm{aggr}} > 0.5$ to exclude the threshold behaviour described in Sec. 3.1.

The dominance of aggregation is hypothesized to originate from the non-linearity in its parameterization. In contrast to the other processes, the conversion rate of aggregation has a squared dependency on the cloud ice content (see Lohmann and Roeckner (1996)), increasing feedback effects between the two.

Additional reasons for the large role of aggregation may lie in its role as a tuning parameter in ECHAM-HAM. For tuning,
uncertain parameters of the model are used (Neubauer et al., 2019). Historically, the scaling factor for the stratiform snow formation rate by aggregation, $\gamma_s$, has been used as it represents a counterpart to the scaling factor for the stratiform rain formation rate by autoconversion. To reach the tuning goals as detailed in Neubauer et al. (2019), it is brought to unrealistically high values (see Table A1). This enhances the changes inflicted by phasing aggregation in this study using $\eta_{\mathrm{aggr}}$. Additionally, structural problems in the model may enhance the role of aggregation artificially. For example, by accounting for heterogeneous
nucleation in the cirrus scheme, which increased ice crystal sizes, Gasparini et al. (2018) were able to reduce $\gamma_s$ by an order of magnitude compared to the reference ECHAM-HAM version (personal communication). This in turn would be expected to reduce the importance of aggregation in the present analysis. Moreover, also the design choices of the CMP scheme, e.g. the order in which processes are called, may influence the results. However, learning about the properties of CMP processes in the ECHAM-HAM model is important, no matter whether they are physically based or artificially introduced through model
design.

A caveat to these results is of course that only CMP processes were investigated here. Parameters or processes from other parts of the climate model as e.g. the dynamics might exhibit an even larger influence on the investigated model output if they were allowed to be varied. For example, Wellmann et al. (2020), using idealized COSMO simulations, found that environmental conditions are more influential for the diabatic heating rates than microphysical processes. However, for the research question at hand, namely how accurate the representation of these four processes within the CMPs needs to be, the comparison of the processes between each other is sufficient. Indeed, the negligible sensitivity of model output to variations in accretion and self-collection of ice suggests that their representation may be simplified (Lee et al., 2012). Contrarily, for an accurate representation of IWP and LWP, the representation of aggregation and riming is where the ECHAM-HAM model developers' attention needs to focus.

### 3.4 Scale dependency analysis

The analysis of global annual mean values yields clear conclusions, but climate models not only need to simulate global mean values correctly but also their spatial and temporal evolution. Since the emulation and subsequent analysis of grid-point level data is tedious and error-prone, we compress the information in the data to a space of lower dimensionality. Choosing to reduce the dimensionality but still represent the whole global data rather than picking certain regions allows for a more objectified and unbiased analysis. This is similar to Holden et al. (2015) who also reduce their high-dimensionality output, albeit with singular value decomposition and Ryan et al. (2018) who use principal component analysis. However, as the model data is complete and on a sphere, a spherical harmonics expansion is our method of choice.

Mathemtically, the model data can be represented as a linear combination of the orthogonal spherical harmonics basis functions as follows:

$$f(\theta,\phi) = \sum_{l=0}^{\infty} \sum_{m=-l}^{l} f_l^m Y_l^m(\theta,\phi) \tag{5}$$

The data $f$ is then a function of the longitude $\theta$ and latitude $\phi$, with $Y_l^m$ a spherical harmonics function of degree $l$ and order $m$ ($l$ and $m$ are integers, with $-l \leq m \leq l$). The complex coefficients $f_l^m$ can be computed as:

$$f_l^m = \int_\Omega f(\theta,\phi) Y_l^m(\theta,\phi) \mathrm{d}\Omega \tag{6}$$

The coefficients make up the angular power spectrum $S_{ff}$:

$$S_{ff}(l) = \frac{1}{4\pi} \sum_{m=-l}^{l} |f_l^m|^2 \tag{7}$$

where the sum over the angular power spectrum $\sum_{l=0}^{\infty} S_{ff}(l)$ is the variance of the data. In principle, an expansion up to order 95 would represent the model data at its resolution of 96 latitudional and 192 longitudional points perfectly, as they are equidistant in spherical coordinates. In practice, we truncated the expansion at the degree $l$ where it represents 95% of the total data variance $\sum_{l=0}^{\infty} S_{ff}$. Thereby we represent the data with as few as possible but as many as necessary basis functions. Note



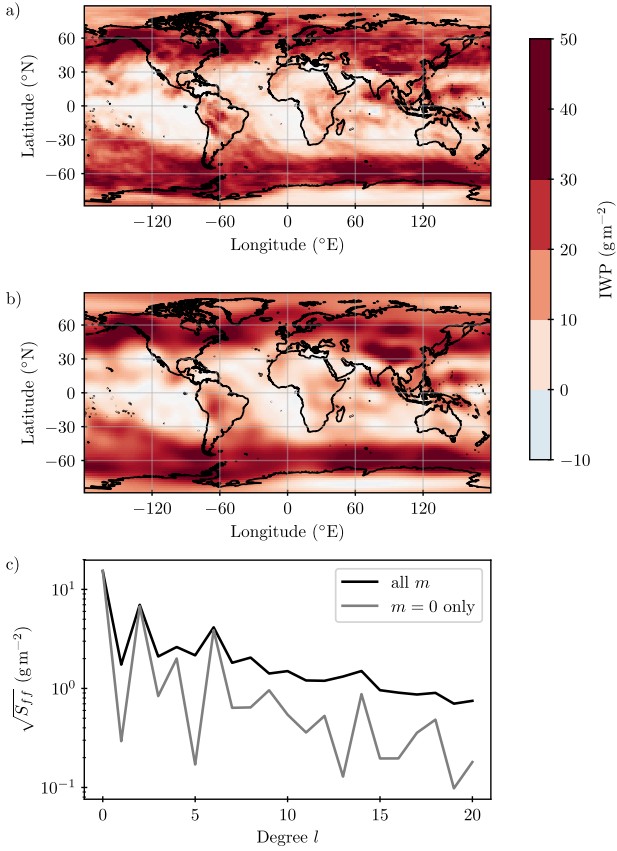

**Figure 9.** Spherical harmonics expansions for one illustrative PPE member ($\eta_{\mathrm{aggr}} \approx 0.31$, $\eta_{\mathrm{accr}} \approx 0.90$, $\eta_{\mathrm{rime}} \approx 0.89$, $\eta_{\mathrm{sci}} \approx 0.36$). **a)** Difference to control ($\eta_i = 1 \forall i$) in the global annual mean IWP, **b)** expansion of spherical harmonics representing the same data as **a)**, generated from the coefficients of the expansion displayed as an angular amplitude spectrum in **c)** as a function of the degree $l$ (with $m$ independent solutions, where modes of $m = 0$ most strongly resemble rotationally symmetric physical patterns of the Earth system such as a North-South contrast). Note that the variability explained by each degree $l$ in general decreases with increasing $l$, which allows us to truncated the expansion at the degree $l$ where it represents 95% of the total data variance.

that in principle, a principle component analysis could yield the same representation with fewer basis functions. However, these functions would depend on the investigated dataset, while the use of spherical harmonics allows for inter-comparability.

Fig. 9 illustrates that a spherical harmonics expansion of the data can serve as an accurate representation, while all the information can be stored in the coefficients up to $l = 20$ instead of on the global grid (see Fig. 9c). Thus confident that the expansion represents the data accurately we can conduct a spatially resolved sensitivity analysis in the spherical harmonics
space. For each variable and degree $l$ a separate emulator was trained on the angular amplitude spectrum $\sqrt{S_{ff}}$, from which samples were drawn as input to the sensitivity analysis. Note that in contrast to the global sensitivity analysis (Fig. 8), all PPE members were used ($0 < \eta_{\mathrm{aggr}} < 2$). As input for the spherical harmonics expansion and subsequent analysis the difference



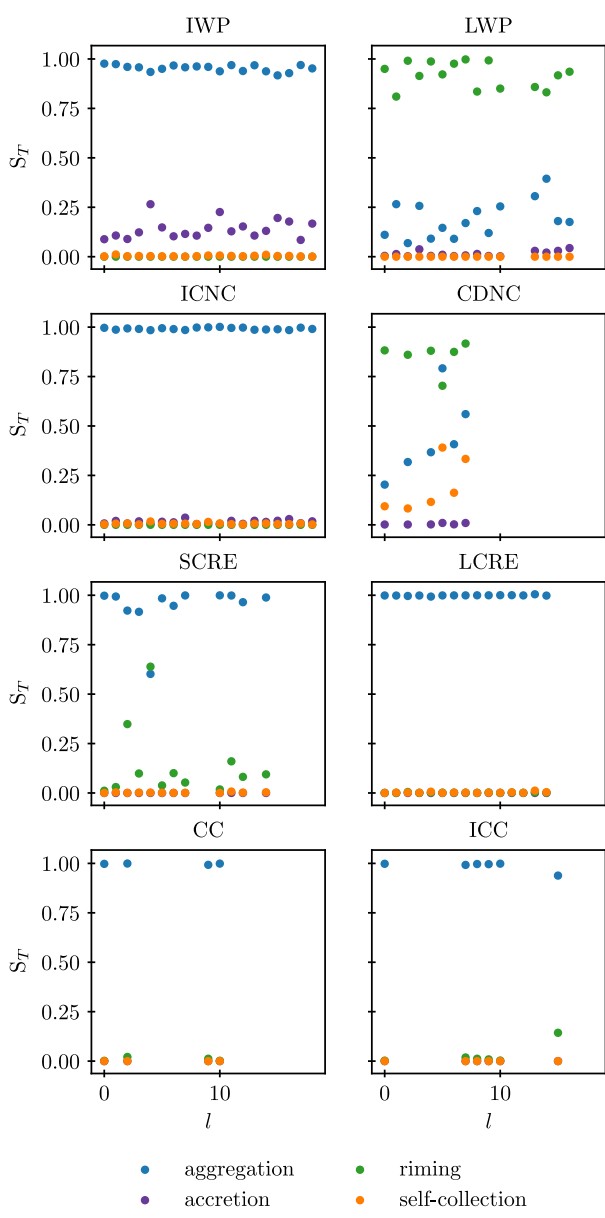

**Figure 10.** Total sensitivity indices for the emulated angular amplitude spectrum as a function of the spherical harmonics degree $l$ for the variables as described for Fig. 8. Note that in contrast to the global sensitivity analysis (see Fig. 8), all PPE members were used ($0 < \eta_{\mathrm{aggr}} < 2$) and the data used was the difference between the PPE members and control simulation ($\eta_i = 1 \forall i$). As detailed in the text, emulators that were found to be defaulting in the validation procedure were not subjected to the sensitivity analysis so that the results for those $l$ is missing here.





of each PPE member to the control simulation ($\eta_i = 1 \forall i$) was used. The validation procedure was the same as described in Sec. 2.4. However, the liquid and mixed-phase cloud cover as well as the total precipitation were excluded as their variations
were too small to be sensibly emulated. Similarly, spherical harmonics members of degree $l$ were excluded from the sensitivity analysis when the emulator was found to be defaulting to an equal prediction over the phase space (see Appendix D). This was the case mostly for degrees $l$ for which the coefficients could be seen to have less amplitude in the angular amplitude spectrum already.

The results are displayed in Fig. 10. For those variables that had total sensitivity indices for aggregation of over 0.7 (IWP,
longwave and shortwave cloud radiative effect, cloud cover, and ICNC) the dominant effect of aggregation is present on all length scales. Accretion is of secondary importance for the IWP, as indicated by the global sensitivity analysis. The CDNC is dominated by riming on all regional scales and on the global scale, while the LWP at some degrees $l < 7$ is also influenced by aggregation.

The emulated surfaces for the spherical harmonics are more uncertain than those for the global mean values (see Appendix
D). This is expected as the training data is more noisy and indicates a less detectable signal on smaller length scales than on the global one. In addition the separate emulation for different degrees $l$ ignores correlations between signals included in multiple degrees $l$, which may lead to the loss of signals that are small in the different $l$ but correlated and therefore should be addressed in future studies. However, as the results of the sensitivity analysis are clear in that variability is dominated by aggregation (see Fig. 10), we can conclude that the results of the global sensitivity analysis also hold on regional scales.

Finally, this analysis demonstrates that spherical harmonics expansion is a viable tool to evaluate model output on all length scales in an efficient and objective manner. Future studies may use it to compare results e.g. from different models. As most expansion degrees are physically difficult to interpret, the method may be expanded to use physically meaningful modes such as the land-sea contrast instead.

### 3.5 Seasonal analysis

Similar to a regional analysis, we use a temporally resolved sensitivity analysis to address the concern that conclusions drawn from annual mean values might not hold on a seasonal scale. Fig. 11 shows the results of the same sensitivity analysis as in Fig. 8, but split by seasons (one emulator per variable was trained and validated for each season; note that in few cases only 47 PPE members were used as with the 48th member the computational constraint was too tight for the emulator). It reveals that indeed the sensitivities to process perturbations are much the same as for the annual mean analysis. This confirms that the conclusions
drawn for model simplifications also hold on a seasonal scale. The model is not sensitive to accretion and self-collection of ice and therefore these processes can be simplified, while aggregation and riming dominate the model response and therefore need to be represented accurately.

### 3.6 Process costs and implications for simplification

The previous analysis shows that the response of ECHAM-HAM to an inhibition of self-collection or accretion is negligible,
while for riming and aggregation at least a less accurate representation can be appropriate. A potential benefit could lie in the





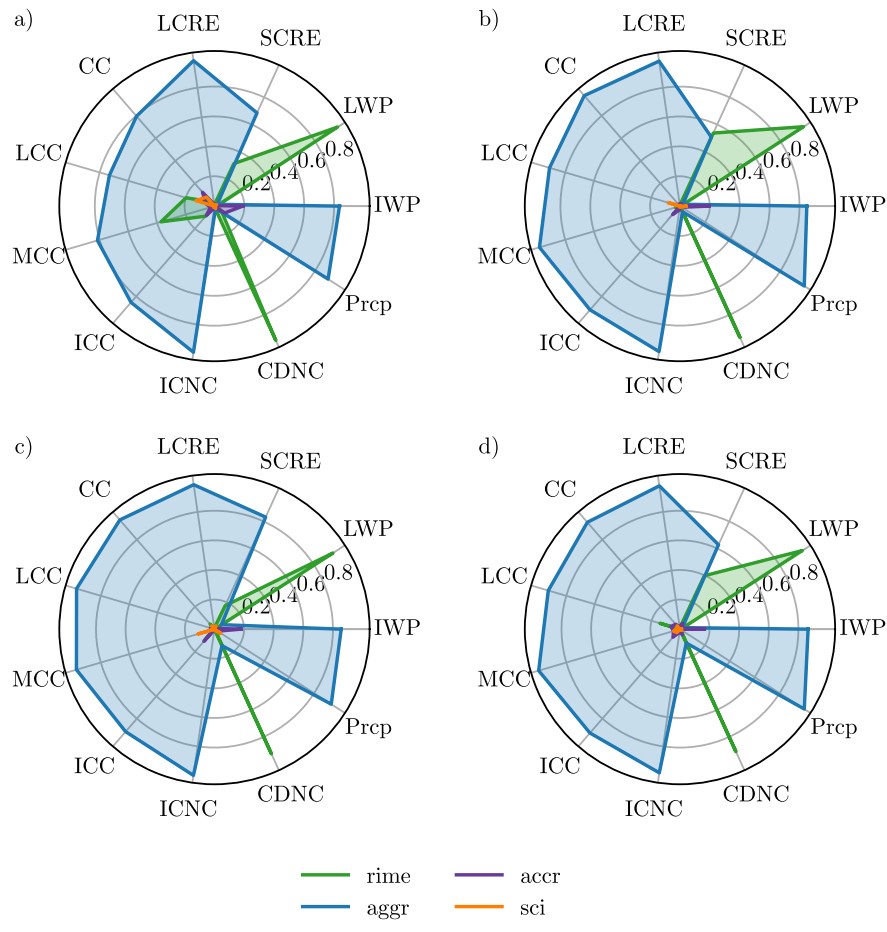

**Figure 11.** Same as Fig. 8 but with seasonal means (**a)** DJF, **b)** MAM, **c)** JJA, **d)** SON) and only total sensitivity indices shown.

reduction of CPU time per model simulation. Table 1 lists the CPU time spent in the CMP routines of the four processes. The timings represent an estimate of how much time could be gained by removing a process from the model. They show that at most, with naively removing (the most drastic simplification) the whole cold precipitation formation routine, only about 0.2% of total computing time can be saved. In a 10 year simulation this would allow for one additional week of simulation, which is

negligible in comparison to the computing needs of e.g. increases in model resolution.

Within the CMP routine there are other physical processes that take up time, but also the calculation of diagnostics and preparational calculations contribute. Of course, if numerous CMP processes and interactions with aerosols were simplified, this would allow for more drastic steps such as fewer aerosol tracers as those could become redundant. Subsequently, significant reductions in model cost could be achieved. Yet by itself, the isolated removal or simplification of CMP processes provides

small leverage for a decrease in computing time. However, as detailed in Sec. 1, there are numerous benefits in simplification that are independent of the associated computing cost, such as a gain in comprehensiveness and interpretability.





**Table 1.** Share of the computing time taken up by the cloud microphysical processes investigated here. In turn, the CMP computing time represents $4.7\%$ of total computing time (excluding diagnostics; all values averaged over a 12-months control simulation with $\eta_i = 1 \forall i$). The time in the subroutine cold precipitation formation that is not attributed to the four processes is used for common initialisations and subsequent processing.

| Process | Share of CMP routine cost (%) |
| --- | --- |
| Riming | 1.8 |
| Aggregation | 0.62 |
| Accretion | 0.46 |
| Self-collection | 0.046 |
| Subroutine cold precipitation formation | 4.8 |

## 4  Summary, conclusions and outlook

This study conducted a sensitivity analysis with an emulated PPE to illuminate the impact of selected CMP processes on model output. Different from previous studies (e.g. Wellmann et al. (2020); Hawker et al. (2021b)) we phase the four CMP processes
of aggregation, riming, accretion and self-collection of ice as a whole. This is achieved by multiplying their immediate effects with a factor between 0 and 2. The resulting response surface of model output and its deviation from results with the default setup serves as a proxy for how accurately a process needs to be represented.

Phasing only one process at a time reveals that ice crystal aggregation acts as a threshold process: phasing it causes the model to deviate, but when it is turned off the deviation is immense. This is because it is the only process that converts ice crystals
to snow and as such accretion and riming depend on it. Using only roughly twice as many simulations than in the one-at-a-time phasing to generate a PPE, we can generate the whole response surface using Gaussian process emulation. A sensitivity analysis of global and seasonal annual means reveals that for cloud cover, ice water path and number concentration as well as shortwave and longwave radiative effect, the phasing of aggregation has the most dominant impact by far. Accretion and riming assume a secondary role. As riming is the only investigated process that directly affects the liquid phase, riming has a
dominant effect on the liquid water path and cloud droplet number concentration. Self-collection of ice has a negligible impact on the investigated global annual mean variables. Resolving smaller horizontal scales using a spherical harmonics expansion of the output variables corroborates the results of the global annual mean analysis, as does a seasonal analysis. These results as well as the shape of the response surface suggest that the parameterisation of self-collection and accretion can be readily and drastically simplified. While aggregation and riming have a large impact on the model output, the shallow slope of the response
surface around the default $\eta_i = 1$ hints that slight modifications of their representations may leave the model output unchanged. The strength of the PPE approach is that interactions are already taken into account, meaning that all four processes could be simplified at the same time. If one wants to make the representation of one of these processes more accurate, aggregation





should be the process of choice as it has the largest leverage in the model and therefore the largest need to be represented correctly.

As we find that the processes themselves use a negligible fraction of the overall model computing time, simplifications are proposed as a means to make the model more interpretable, not cheaper (see Sec. 1 and 3.6). Our analysis shows that the representation of the four investigated microphysical processes leaves room for simplification. At the least, when new parameterisations are included in climate models we should question their implementation also regarding the complexity they add, looking for their consistency, interpretability, simplicity and comprehensiveness (Mülmenstädt and Feingold, 2018;

Touzé-Pfeiffer et al., 2021).

This study introduces the methodological framework to study the sensitivity of a climate model to the representation of CMP processes. To complete it, the analysis needs to be expanded to include other CMP processes in the model: For cold CMP ice formation, regional modeling studies have demonstrated cloud susceptibility to the choice of the ice nucleation parameterisation (Levkov et al., 1995; Hawker et al., 2021b), whereas in ECHAM-HAM heterogeneous immersion freezing

in mixed-phase clouds has been shown to be rather inefficient (Villanueva et al., 2021). More generally the heterogeneous ice formation pathway in mixed-phase clouds is small in ECHAM-HAM (Dietlicher et al., 2019; Bacer et al., 2021), hinting at simplification potential. In a sensitivity study of CMP parameters, Tan and Storelvmo (2016) found that the time scale of the Wegener-Bergeron-Findeisen process explains a large variance in supercooled cloud fractions, suggesting that as a whole it may be a dominating process as well. Secondary ice formation (Korolev and Leisner, 2020) may interact with the ice crystal

source processes, allowing for interactive sensitivities (Hawker et al., 2021b), and should therefore be included, even though only the Hallet-Mossop process is optionally included in ECHAM-HAM (Neubauer et al., 2019). Moreover, for a complete CMP process investigation, of course the warm rain processes need to be included as well (Wood et al., 2009; Gettelman et al., 2013)).

One might argue that our analysis neglects the influence of other factors external to the CMPs on our conclusions. However,

as our simulations span the whole globe and a whole year, they cover a range of dynamical situations and the results are therefore robust in the current climate. Whether the conclusions hold e.g. in a future changed climate will have to be evaluated in a future study. It is important to stress that while we propose that simplifications to the CMP representation are possible, care needs to be taken to leave them physically based to ensure that the model can correctly represent differing climates. Another factor that has not been investigated here is the model resolution that may affect the CMP behaviour in the model and thereby

our conclusions on single processes' importance (Santos et al., 2021). The implementation and design choices of the CMP scheme in ECHAM-HAM may also influence the results, e.g. in the order of processes that are called, as well as the employed tuning strategy.

Nevertheless, learning about the representation of CMP process in ECHAM-HAM and how sensitive the model is to their representation helps us to interpret and improve the model, especially when comparing the results to experimental studies. To

this end, it will also be fruitful to compare our findings to sensitivities in other models using different CMP schemes.



*Code and data availability.* The ECHAM-HAMMOZ model is freely available to the scientific community under the HAMMOZ Software License Agreement, which defines the conditions under which the model can be used. The specific version of the code used for this study is archived in the ECHAM-HAMMOZ SVN repository at /root/echam6-hammoz/tags/papers/2021/Proske_et_al_2021_ACPD. More information can be found on the HAMMOZ website (https://redmine.hammoz.ethz.ch/projects/hammoz, last access: 17 September 2021).

Analysis and plotting scripts are archived at https://doi.org/10.5281/zenodo.5506588 (Proske et al., 2021a). Generated data is archived at https://doi.org/10.5281/zenodo.5506533 (Proske et al., 2021b). The PyDOE library (tisimst, 2021) was used for Latin Hypercube Sampling, GCEm (Watson-Parris, 2021; Watson-Parris et al., 2021) for the construction of the emulator, SALib (Usher et al., 2020) for the sensitivity analysis, and PySphereX (Staab, 2021) for the construction of the spherical harmonics expansion.





# Appendix A: Tuning

**Table A1.** Tuning parameters that differ between this study and the reference of Neubauer et al. (2019). $\gamma_r$ is the scaling factor for the stratiform rain formation rate by autoconversion. $\gamma_s$ is a scaling factor for the stratiform snow formation rate by aggregation. With the changes described in Sec. 2.1 the tuning parameter of the maximum cloud droplet radius, $r_{CDNC}$, replaces the previous minimum cloud droplet number concentration, $CDNC_{min}$. The tuning parameter for immediate aggregation of detrained ICNC, $\gamma_d$, is newly introduced.

| Parameter | ECHAM-HAM this study | reference |
|:---:|:---:|:---|
| $\gamma_r$ | 5 | 10.6 |
| $\gamma_s$ | 600 | 900 |
| $r_{CDNC}$ | $15 \cdot 10^{-6} \, \mathrm{m}$ | – |
| $CDNC_{min}$ | – | $40 \cdot 10^{-6} \, \mathrm{m}^{-3}$ |
| $\gamma_d$ | 5 | – |





## 525 Appendix B: PPE results for more variables

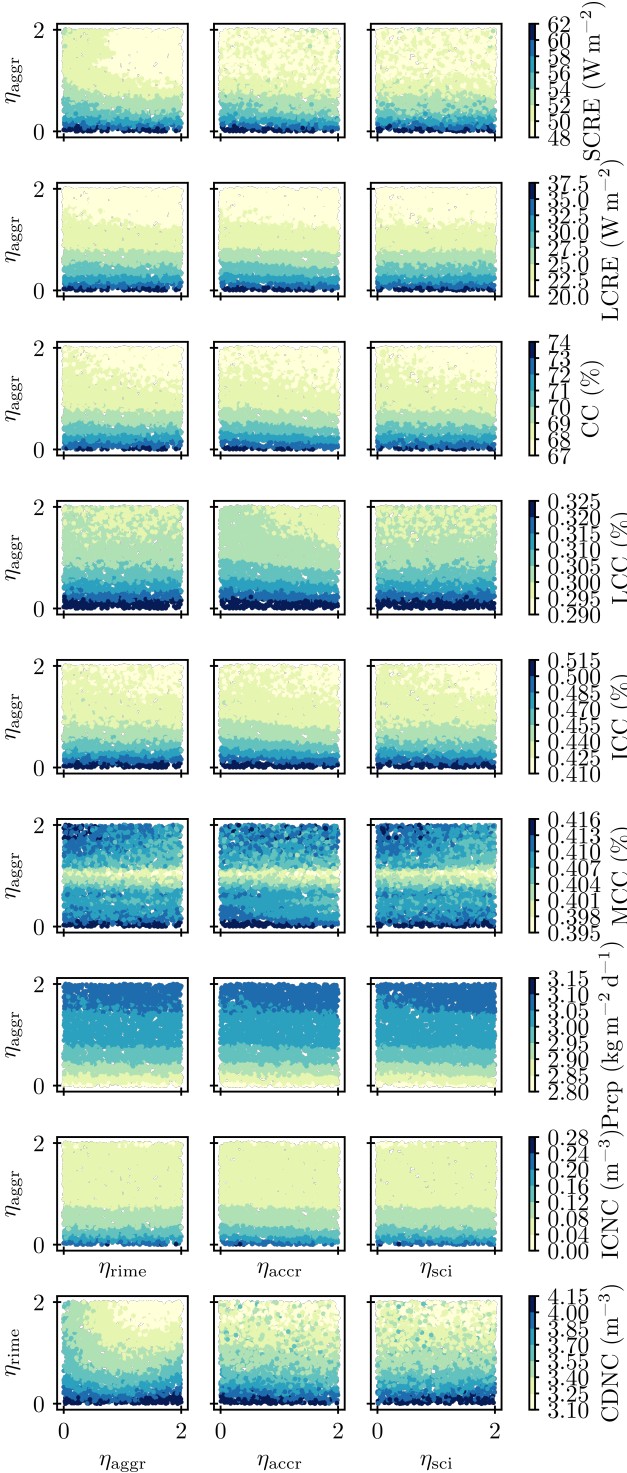

**Figure B1.** Visualisation of the multi-dimensional response surfaces of the emulated PPEs for multiple variables. Each process is a dimension, and the colorbars denote the global annual mean values. In principle, each surface could be displayed by a full matrix plot as in Fig. 6 and 7, but here only the panels that include the dominating process are shown (aggregation, except for CDNC in the last row, where riming is the dominant process).





## Appendix C: Sensitivity of the sensitivity analysis to constraining to $\eta_{\text{aggr}} > 0.5$

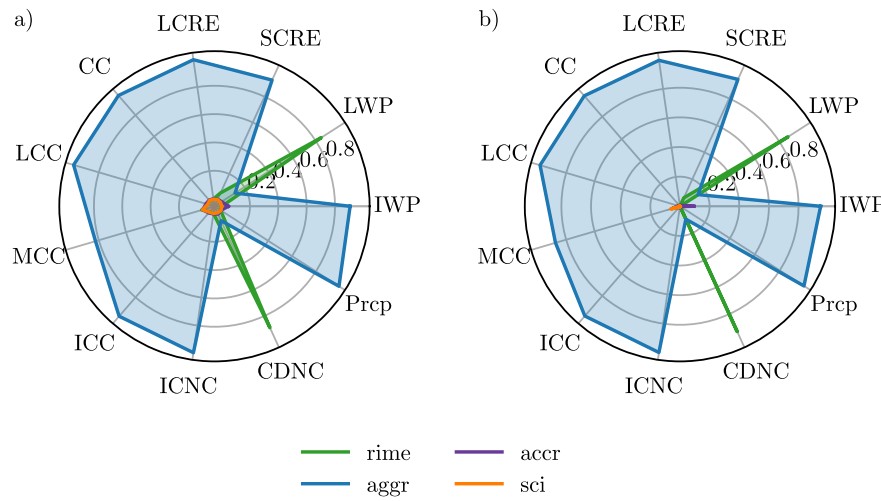

**Figure C1.** Same as Fig. 8 but using the whole response surface for the sensitivity analysis.



## Appendix D:  Validation of the spherical harmonics sensitivity analysis

The validation of the spherical harmonics emulation was carried out as described in Sec. 2.4. Larger uncertainties in the emulation were apparent for almost all variables and degrees $l$ (see Fig. D1 for an example) than for that of the global mean values. However, some emulations were also found to be defaulting, meaning that they predicted a similar output value for the whole phase space (see Fig. D2 and D3 for an example). As this behaviour points to a missing signal in the input, these points were excluded from further analysis, if the following two criteria were not fulfilled:

- The uncertainty in the prediction is smaller than the spread of the variable, i.e. the smallest error bar in Fig. D1d) is smaller than $0.9\Delta Y_{\mathrm{sim}}$.

- The predictions are significantly different from each other, i.e. there is one pair of predictions whose error bars do not overlap.

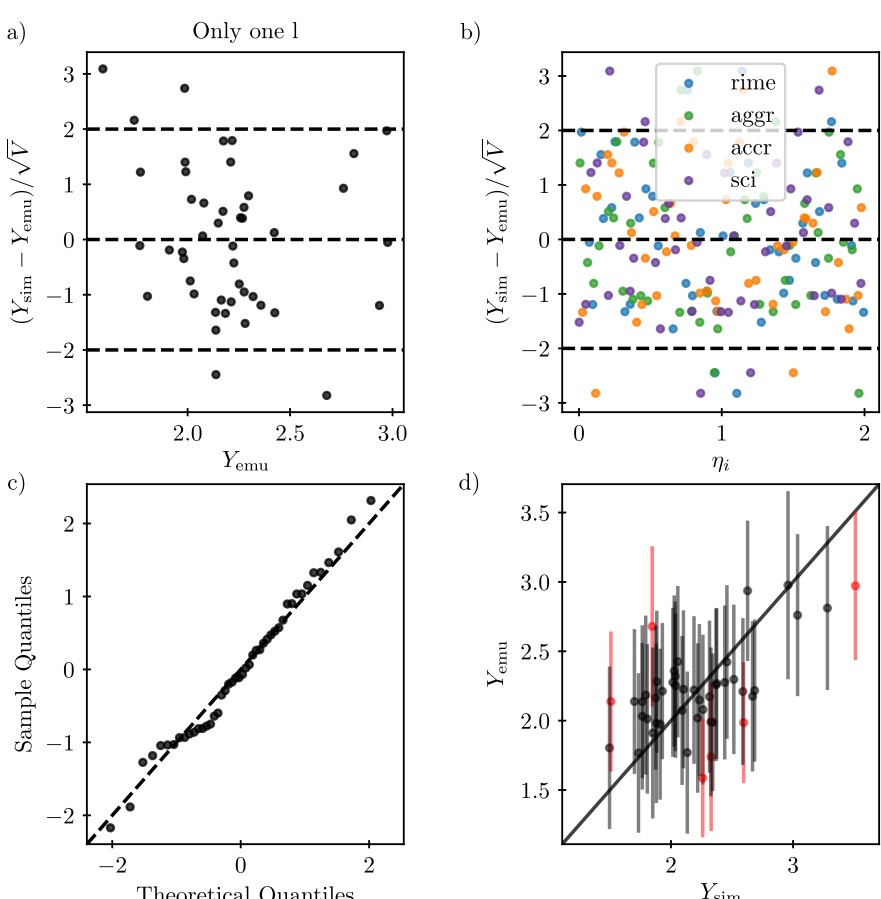

**Figure D1.** Validation of the emulated angular amplitude spectrum of degree $l = 16$ for the LWP.



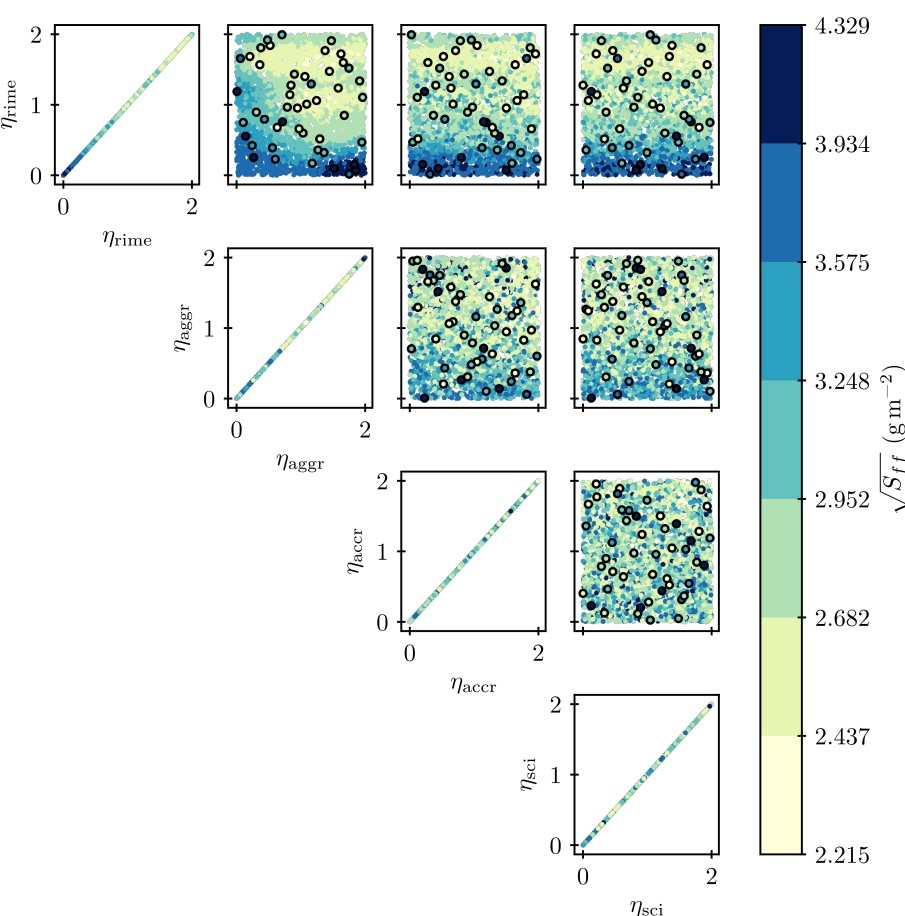

**Figure D2.** Same as Fig. 6 but for the LWP spherical expansion angular amplitude spectrum of degree $l = 11$. In this case, the emulator was found to be defaulting and therefore failed the validation and was not included in the subsequent sensitivity analysis. The points enclosed by black circles denote the PPE member results used to train the emulator.

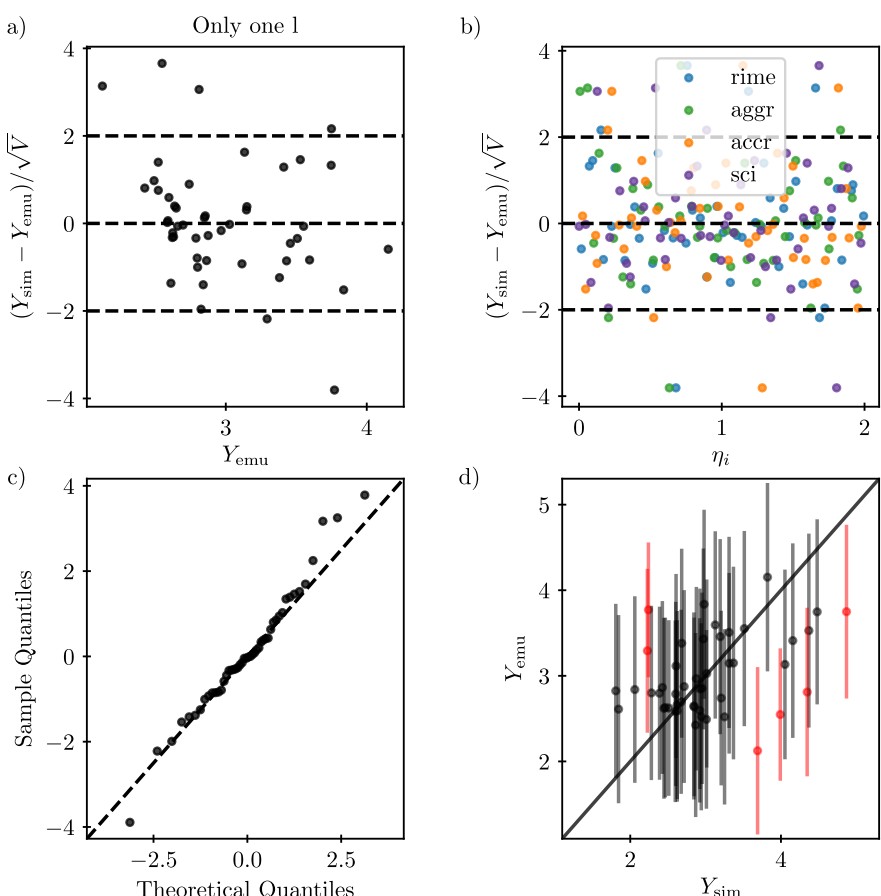

**Figure D3.** Validation of the emulated angular amplitude spectrum of degree $l = 11$ for the LWP (see Fig. D2), which failed because of diagnosed defaulting.



*Author contributions.* UP developed the model code, ran the simulations and emulation, analysed the data and wrote the manuscript. UP, SF, DN and UL developed the study idea and design and analysed the results. MS and UP developed the idea for the spherical harmonics analysis, for which MS developed the code. SF, DN, UL and MS edited the manuscript.

*Competing interests.* The authors declare no conflict of interest.

*Acknowledgements.* The authors thank Duncan Watson-Parris for his advice on using GCEm and helpful discussions. They are grateful to to Rachel Hawker and Leighton Regayre for their advice on the emulation.

Throughout this study, the programming languages CDO (Schulzweida, 2018) and Python (Python Software Foundation, www.python.org) were used to handle data and analyse it. This project has received funding from the European Union's Horizon 2020 research and innovation

programme under grant agreement No 821205 (FORCeS).





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
