# Peer review of "Assessing the potential for simplification in global climate model cloud microphysics"

_Atmospheric Chemistry and Physics, 2021_

## Author Comment (AC1)

**Author Response to Reviews of**

**Assessing the potential for simplification in global climate model cloud microphysics**

Ulrike Proske, Sylvaine Ferrachat, David Neubauer, Martin Staab, and Ulrike Lohmann

*Atmospheric Chemistry and Physics,* `doi:10.5194/acp-2021-801`
* * *
RC: *Reviewer Comment*,    AR: *Author Response*,    ☐ Manuscript text

We sincerely thank the reviewers for their insightful and constructive feedback. We implemented their feedback into a revised version of the manuscript. Please find our answer to the reviewers' points below, followed by a marked-up manuscript verison.

**1. Reviewer Comment #1**

RC: *This study used a perturbed parameter ensemble (PPE) framework together with Gaussian process emulation to explore sensitivity of a global climate model to perturbations of four ice microphysical processes. The paper first describes analysis of global results, then a spatial decomposition using spherical harmonics expansion. Overall results show strong sensitivity to the "aggregation" process, while riming has a strong impact on LWP. The authors then discuss implications for the level of complexity of microphysical process representations. Overall, the paper is well written and addresses an important topic, and is within the scope of Atmospheric Chemistry and Physics. This study is interesting. The methodology appears to be sound. The demonstration of this PPE-based approach to investigate cloud microphysical processes as a "proof of concept" is the strongest part of the paper in my opinion. However, I do have several questions regarding the broader context, including the motivation for this approach and interpretation of results. Overall I think it could be acceptable in ACP, but more context is needed and some clarification of points raised below. These are detailed in "specific major comments". I also have a several additional minor comments and a few editorial corrections.*

AR: *Thank you for your clear and constructive feedback. Please find our respective answers directly below your comments below.*

**1.1. Specific major comments**

**1.1.1**

RC: *I don't think "phasing" is the best term to use for the process perturbations. Essentially, you're applying a multiplicative factor directly to the process rates. But "phasing" typically means the relationship between the timing of two or more events, or synchronizing of multiple events. Similarly, the paper also uses the term "phase in and out" which implies some kind of dynamic or periodic change to the multiplicative factors, which is not the case here (they are constant in time and space for a given run, as I understand). Instead, process "perturbing" seems like a more appropriate term than "phasing". I don't think addressing this is required before publication, but I strongly suggest not calling this "phasing" to avoid confusion.*

AR:   *To avoid confusion, we have followed your suggestion and use "perturbing" instead of "phasing" except where the dynamic connotation of "phase in and out" is intended.*

**1.1.2**

RC:   ***My main comment, as noted above, is in the interpretation and broader context of the study. The authors motivate this study by suggesting that results from this approach can be used to inform which processes can be simplified, but the specifics of this are unclear and the paper seems self-contradictory in several places. For example, the argument on lines 320-325 is: "Recognizing it as a threshold process and seeing the gradual response to small deviations from 1.0 in $\eta_{\mathrm{aggr}}$, it appears that there is potential for a less accurate description of aggregation in the model.", where "it" in this sentence means the "aggregation" process. I don't understand this argument. Moreover, this seems to directly contradict the argument later on lines 385-389 and lines 445-447. These lines state that the model is sensitive to representation of "aggregation" and riming, and therefore these processes "need to be represented accurately" and should receive attention of model developers, while self-collection and accretion can be simplified. But then this is contradicted again on line 450: "The previous analysis shows that the response of ECHAM-HAM to an inhibition of self-collection or accretion is negligible, while for riming and aggregation at least a less accurate representation can be appropriate." Perhaps this is not what the authors intended to write and it's simply written incorrectly? Either way, this is confusing. See also lines 478-479 in the conclusions section, and lines 481-484, where the latter again argues that aggregation should be the "process of choice" to represent most accurately. Overall, these self-contradictory arguments make the paper feel somewhat incoherent, and this needs to be addressed before the paper can be accepted.***

AR:   *We see that in the presented form our argument was confusing. Our argument is indeed going two ways: One the one hand aggregation is the dominant CMP process of the four investigated. As such, if one wanted to improve the model, aggregation provides leverage to doing so. This means that the representation of aggregation should be subjected to scrutiny and - as you point out in other comments - that its dominant role might need to be reconsidered (and if it is considered unphysical, one solution might be to move to a scheme that does not separate the ice and snow categories). On the other hand, around $\eta_{\mathrm{aggr}} = 1$ the considered model output variables are unresponsive to a change in $\eta_{\mathrm{aggr}}$. This implies that a change in the formulation of aggregation would keep the model intact, and thus simplifications in the formulation of aggregation are possible as well. We have reformulated the sentences you point to to make this argument clearer.*

> The implications for possible simplifications are different: seeing only the large difference between a simulation with and without autoconversion, one would think that this is an immensely important process. Recognizing it as a threshold process and seeing the gradual response to small deviations from 1.0 in $\eta_{\mathrm{autc}}$ (similar to the purple curve in Fig. 2), it appears that there is potential for a less accurate description of  autoconversion in the model.

Indeed, the negligible sensitivity of model output to variations in accretion and self-collection of ice suggests that their representation may be simplified (Lee et al., 2012).  Due to the small deviations in the considered variables in response to variations around $\eta_i = 1$ for riming and autoconversion (purple line in Fig. 2), there is potential for slight simplifications of their formulations. In the grand scheme of CMP parameterization development, however, autoconversion as the most dominant process of  four is a key process to scrutinize given the possibly troubling origin of this dominance in its role as a tuning factor.
* * *
The model is not sensitive to accretion and self-collection of ice and therefore these processes can be simplified, while  autoconversion and riming dominate the model response .
* * *
The previous analysis shows that the response of ECHAM-HAM to an inhibition of self-collection or accretion is negligible, while for riming and  autoconversion a less accurate representation can be appropriate.
* * *
These results as well as the shape of the response surface suggest that the parameterisation of self-collection and accretion can be readily and drastically simplified. While  autoconversion and riming have a large impact on the model output considering the whole investigated phase space, the shallow slope of the response surface around the default $\eta_i = 1$ hints that slight modifications of their representations may leave the model output unchanged. The strength of the PPE approach is that interactions are already taken into account, meaning that all four processes could be simplified at the same time. If one wants to  develop the CMP scheme further, autoconversion is the process to scrutinize as it has the largest leverage in the model and therefore  most urgent need to be represented correctly.

**1.1.3**

RC: *I have several other questions and comments about the broader motivation and context for this study. First, a process could still be important even if there is not much sensitivity to some limited set of output metrics in a given climate state (for example, a process might be important for cloud-aerosol interactions even if the impact on LWP, IWP, etc for the current climate state is small). This is alluded to on lines 507-508, but I feel this point should be emphasized more in the paper (e.g. in the introduction). Moreover, if a process is unimportant, it doesn't necessarily "do harm", except for extra computational cost. Computational cost here is negligible for these processes anyway, as discussed on lines 451-455 and Table 1. I appreciate the gain in interpretability with model simplification (stated explicitly on line 460), and this is an important point, but this should be made clear in the introduction when motivating the work.*

AR: *We have added a sentence about the transferability of simplifications into other climate states in the introduction:*

> Please note that the potential for simplification is evaluated in the current climate. Thus any derived simplifications would need to be evaluated against a reference model for their suitability in a changed climate state prior to employing it in e.g. climate change projections.

AR: *However, we believe that the introduction already gives a fair view of the gain in interpretability with simplifications to motivate this work. Furthermore, it explicitly mentions the dangers of detailing an unimportant process as making the model more difficult to understand, adding to the difficulty of model intercomparisons and overinterpreting those processes that are represented in detail while neglecting those that are not.*

RC: **Also, regarding the sentence on line 460, I don't see how such simplification leads to a gain in "comprehensiveness", which I assume means generality; could you clarify this?**

AR: *We corrected our incorrect choice of words for "comprehensiveness" by replacing it:*

> However, as detailed in Sec. 1, there are numerous benefits in simplification that are independent of the associated computing cost, such as a gain in  compactness, robustness and interpretability.

RC: **Furthermore, regarding both computational cost and interpretability (as well as comprehensiveness), it seems likely that other scheme features, particularly the choice of prognostic variables and hydrometeor categories, is more important than process formulations anyway (see major comment #4 below for further discussion of this point).**

AR: *This point about the other scheme features is answered with #4 below.*

**1.1.4**

RC: **The process formulations themselves only one aspect of scheme complexity, and arguably a rather small part. The choice of hydrometeor categories and prognostic variables is generally more important in terms of determining overall scheme complexity, interpretability, comprehensiveness, and computational cost. Yet nothing is mentioned about this in the paper. In my experience, simplifying the representation of microphysics (ice microphysics in particular) in terms of choice of prognostic variables and hydrometeor categories is likely to have a \*much\* larger effect on interpretability, cost, etc. than only simplifying the process formulations. To this end, can your approach inform simplification of these other aspects that are likely to be more important overall? As a specific example, one of the main findings of this paper is the importance of "aggregation" (which is probably better termed "ice autoconversion", see major comment #7 below). Yet in simpler schemes that use a single ice category but multiple predicted properties for that category (e.g., Morrison and Milbrandt 2015; Eidhammer et al. 2017) this process is not even included, nor are any other conversion processes between ice categories (e.g. accretion of cloud ice by snow). The critical point is that the separation of self-collection, aggregation, and accretion using separate cloud ice and snow categories is artificial, and the approach ultimately must rely on an ad-hoc choice for how cloud ice and snow are separated. Given that "aggregation" (autoconversion) is the most important process (of the 4 tested) based on your results, it's troubling that there is little physical constraint for this process. To me, the findings from this study motivate the kind of simplification of using one ice category so that**

*"aggregation" is no longer needed. I feel discussion of this issue is needed, particularly the approach of moving away from separate categories for cloud ice and snow. More broadly, discussion of these other sources of structural uncertainty (categories, prognostic variables, etc.), besides just the process formulations, is needed in the paper.*

AR: *We have added a discussion of this point into the conclusion section:*

> *Of course, more drastic simplifications than process reformulations would provide more leverage on interpretability and computing cost. For example, CMP schemes that contain only one tracer category for ice, e.g. the Predicted Particle Properties (P3) ice microphysics scheme (e.g. Morrison and Milbrandt (2015), Eidhammer et al. (2017), Dietlicher et al. (2018), Dietlicher et al. (2019), and Tully et al. (202. are more physical as well as more interpretable. From this perspective it might seem troubling that in the current CMP scheme the autoconversion process, which is a transfer mechanism between the two artificial classes, is so dominant in its importance. However, while the categories are artificial, the process itself is not: accretion of ice crystals forming larger ice crystals would be the equivalent process with only one ice category. Still, autoconversion is difficult to constrain in observations (Morrison et al., 2020), and so moving towards a one ice category scheme seems advisable.*

**1.1.5**

RC: *It's not clear what authors mean by aggregation as a "threshold process", since the response to $\eta_{\mathrm{aggr}}$ is smooth and gradual (e.g. in Fig. 4). This "threshold" is mentioned many times throughout the paper. I assume this refers to the model not producing snow when eta = 0 for "aggregation", and thus riming and other snow processes also are zero. If so, please state this clearly in the paper the first time the threshold behavior is mentioned. I also question whether this should be called a threshold. I think of a "threshold" as some value that when crossed leads to a large, abrupt change. Is this really what happens? Turning off a process completely doesn't really seem to quality as a "threshold" from this point of view. Thus, my suggestion is to refer to this as something else besides a threshold.*

AR: *We have added a data point for $\eta_{\mathrm{aggr}} = 0.1$ to illustrate the behaviour we are referring to more clearly. We have also added more detail to the explanation of this behaviour in the text the first time it is mentioned. In lack of a better term, we continue to refer to it as "threshold" behaviour but hope that with the explanation this is now more clear.*

> The shape of the model response to the gradual  perturbation of the processes holds additional information: while the generated model response is mostly gradual, for low  $\eta_{\mathrm{autc}}$ the response is more abrupt. This behaviour, which we call a threshold response, is most striking for the global annual mean LWP, for which the signal for  $\eta_{\mathrm{autc}} \geq 0.25$ is not significantly different to that of accretion and self-collection. When  autoconversion is completely inhibited, the LWP increases dramatically and the signal becomes stronger than that for riming, which had increased consistently and gradually. This behaviour can be explained by  autoconversion acting as a  catalytic process for accretion and riming, creating what we call a threshold behaviour when it is turned off.

RC: *A few other related comments. First, it's not entirely clear that snow should actually be zero when "aggregation" is turned off, if there are other processes which might contribute at least some to snow (see major*

*comment #8 below). Second, this behavior really just reflects assumptions in the scheme, particularly the separation of total ice into cloud ice and snow categories (see major comment #4 above). For example, the shutting off of riming as aggregation goes to 0 must reflect the fact that riming of snow is included but riming of cloud ice is neglected (at least, I think riming of cloud ice is neglected but that isn't clear from the paper). Ultimately this kind of behavior does not necessarily reflect actual underlying cloud physics but rather the separation of ice into cloud ice and snow categories with an ad-hoc conversion between the two called "aggregation" (ice autoconversion).*

AR: *This is correct: with this study we first and foremost investigated the cloud microphysics scheme in ECHAM-HAM. The results reflect the scheme behaviour and we do not intend to infer statements about the significance of the investigated processes in reality. We have added to the manuscript as follows to make this clear:*

> *The implementation and design choices of the CMP scheme in ECHAM-HAM may also influence the results, e.g. in the order of processes that are called, the separation between ice and snow, as well as the employed tuning strategy. Thus the results as such are only applicable to this CMP scheme and cannot be transferred to the significance of the investigated processes in reality.*

**1.1.6**

RC: *I have a few comments on the process perturbations. The range for the process perturbations (magnitude of $\eta_i$) seems ad-hoc, from 0 to 200%. Following the motivation of understanding sensitivity to process rates and the level of process uncertainty, it seems that the range of $\eta_i$ should follow from the level of uncertainty of a given process. However, turning off a process entirely ($\eta_i = 0$) is unrealistic and generally not justified from a physical standpoint or from the standpoint of the actual range of uncertainty of a process. I understand why this might be done as a sensitivity test and "proof of concept" to assess importance of that process, but this point should be clarified in the paper. Also, what determined the upper range of 200%? Was this ad-hoc?*

AR: *In response to yours and the second reviewer's comments, we have adjusted the range of the perturbations to go from $\eta_i = 0.5$ to 2 in the PPE. The choice of going to 200% was ad-hoc, but is now matched by the multiplicatively appropriate lower end of the range. We think that this range is appropriate as we're interested in perturbations around $\eta_i = 1$ (as demonstrated in the newly added sketch) and a deviation of double or half the original results through simplification seems reasonable for all four processes (e.g. doubling corresponds to implicitly adding a new process). The adapted PPE design also makes the "proof of concept" idea more apparent in the one-at-a-time sensitivity studies going until $\eta_i = 0$, conducted for e.g. Fig. 4.*

RC: *Finally, a fairly major simplification of this study is assuming the process perturbations are constant in time and space. Of course, introducing time/space dependence would add more dimensions and increase complexity of the problem considerably, and therefore I can see why constant perturbations were assumed in this study for practical reasons. But from the standpoint of process uncertainty, it seems likely that relative uncertainty in these processes is not uniform in time and space. I suggest briefly mentioning this issue.*

AR: *We have added a mention of this issue as suggested:*

> Note that the perturbations are constant in space and time for each PPE member, serving as a reasonable proxy for the effect of possible simplifications, which would likely be variable in space and time.

**1.1.7**

**RC:** *The term "aggregation" for conversion of cloud ice to snow by ice crystals aggregating with other crystals is confusing, and generally inconsistent with past literature on microphysics schemes. In other schemes this is referred to as "ice autoconversion". "Aggregation" is confusing because "self collection" also occurs by the physical process of aggregation – the same physical process, except that particles remain within the cloud ice category. Note that some other ice autoconversion schemes also account for growth from cloud ice to snow via vapor diffusion (e.g., Harrington 1995), which can be an important process producing large ice particles. I'm assuming that's neglected here, and the "aggregation" (or autoconversion) is only formulated via ice-ice collection? Or in some other way? Overall, I strongly recommend the authors refer to this process as "autoconversion" rather than "aggregation".*

**AR:** *Thanks for this suggestion. We have adapted the naming of the process to autoconversion instead of aggregation (and autc instead of aggr) in the manuscript.*

**1.1.8**

**RC:** *It's mentioned that "aggregation" (autoconversion) is the only process that can generate snow. Is this really true? What about freezing of rain? Does that form cloud ice, or what else is done?*

**AR:** *Indeed aggregation of ice crystals is the only process that can form snow in the model. Rain is diagnosed and precipitates to the surface within one timestep and therefore cannot freeze.*

**1.1.9**

**RC:** *You might consider briefly mentioning other emulation approaches which have some benefits (as well as some drawbacks) compared to using Gaussian processes for emulation as done here, specifically those based on neural networks. I believe some of these issues are outlined in Watson-Parris et al. (2021), and perhaps you could just mention the issue in a quick sentence or two and cite that paper (already cited elsewhere in your paper).*

**AR:** *We have added this note as suggested:*

> We prefer the Gaussian process emulator over e.g. a neural network because of its demonstrated suitability and need for fewer input data (see Watson-Parris et al., 2021b, for a more in-depth discussion).

**1.1.10 Figure 5**

**RC:** *These plots must be showing the column-integrated process rates, but this isn't stated anywhere. Otherwise the units don't make sense.*

AR: *Thanks for noting this, we have added a specification to the caption.*

**1.1.11**

**RC:** *There have been several studies (mainly authored by Posselt, van Lier-Walqui, and/or Morales) that have examined sensitivity to microphysical scheme perturbations in a Bayesian framework. These studies have used MCMC directly rather than emulation, which was possible by focusing on idealized modeling with reduced dimensionality (1D or 2D models generally, or small 3D domains). Some examples are Posselt (2016), Morales et al. (2021), He and Posselt (2015), and van Lier-Walqui et al. (2020) (references given at the end of this review). While these studies haven't focused on global climate, they seem relevant to mention. In particular, the study of van Lier-Walqui et al. (2014) is relevant because they similarly applied constant multiplicative factors to individual microphysical process rates as a way to vary processes and examine the effects of process uncertainty (but in a 1D model, with MCMC).*

AR: *Thank you for pointing us to this set of references which had previously escaped our attention. We agree that they are relevant and have added references in the Introduction and Methods Section of the manuscript:*

> *In a PPE multiple input parameters are perturbed at the same time. In this way, PPEs are expanding upon sensitivity studies that vary one parameter (e.g. Lohmann and Ferrachat (2010) and He and Posselt (2015)) or multiple parameters at a time (e.g. Ghan et al. (2013)), allowing to investigate the interaction effects of perturbations within the whole possible parameter space. For example, Sengupta et al. (2021) used a PPE to determine the impact of parameters related to secondary aerosol formation on organic aerosol in a global aerosol microphysics model. In a next step, parameter ranges can be constrained when comparing the PPE to observations (Posselt, 2016; van Lier-Walqui et al., 2014; van Lier-Walqui et al., 2019): Morales et al. (2021) built a PPE of CMP process parameters and environmental conditions, generated using a Markov Chain Monte Carlo algorithm, in idealized simulations to then constrain the parameters with artificial observations.*

> This perturbation of whole processes was introduced by van Lier-Walqui et al. (2014) to estimate the uncertainty including errors in the physical assumptions of process formulations. In our case, the parameters aid to understand the sensitivity of the model to each process: From the response of model output to variations in $\eta_i$, we can extract information on how accurately a process $i$ needs to be represented in the model.

**1.2. Specific minor comments**

**1.2.1 Line 7**

**RC:** *"phasing of a process" isn't clear and without context most readers will not understand what this means. As argued in major comment #1, I suggest you consider using different terminology than "phasing" altogether.*

**AR:** *As detailed in our answer above, we changed "phasing" to "perturbation".*

**1.2.2 Line 27**

**RC:** *This could give the false impression that Fisher and Koven (2020) concerns cloud microphysics in climate models specifically, when it's about land surface processes. This is because you say "these processes" in the sentence before on line 25, which refers specifically to cloud microphysics. There's also a paper by Morrison et al. (2020) that discusses similar issues as on lines 23-35, but specifically from the standpoint of microphysics.*

**AR:** *Thank you for pointing us to this reference we were previously unaware of. We have adapted the paragraph to add this reference as well as to make clear that Fisher and Koven (2020) concerns land surface modelling:*

> *Responding to the challenge of incorporating these processes in climate models, the community has added more and more processes into GCMs (Knutti and Sedláček, 2013) with increasing detail in their representation (e.g. Archer-Nicholls et al. (2021) and Morrison et al. (2020)). As Fisher and Koven (2020) argue  for the similar situation in land surface modelling, this may be due on the one hand to scientists' tendency to focus on their own area of expertise. On the other hand, it also reflects the fact that the Earth system is indeed complex and that many processes may matter (Morrison et al., 2020). However, it is doubtful whether more detail will help us to reduce uncertainty (Knutti and Sedláček, 2013; Carslaw et al., 2018). More complexity  also has its downsides: More parameterised processes lead to more parametric uncertainty which in turn scientists investigate and try to reduce with large scientific effort (e.g. Rougier et al. (2009), Lee et al. (2011), Yan et al. (2015), Williamson et al. (2015), and Dagon et al. (2020)). In fact, Reddington et al. (2017) argue that "aerosol-climate models are close to becoming an overdetermined system with many interacting sources of uncertainty but a limited range of observations to constrain them",  referring to the complexity in the representation of aerosols and their interaction with clouds. This is related to equifinality, meaning that model versions from different regions of the input parameter space may lead to the same results that compare well with observations. These models may simulate a range of aerosol forcings (Lee et al., 2016), which is not possible to constrain with current observations. Morrison et al. (2020) diagnose the same problem for CMP schemes, whose complexity, they say is ""running ahead" of current cloud physics knowledge and the ability to constrain schemes observationally."*

**1.2.3 Line 66**

**RC:** *I don't think that "stripped of detail" means "less accurate". Accuracy implies how close the model is to some benchmark or truth. Thus, my suggestion is to simply state "...where process parameterisations can be stripped of detail to aid the development of simplified models ...".*

AR: *Thank you for the suggestion. We have adapted the sentence accordingly.*

> In this paper, we propose a new methodology to assess where process parameterisations can be  stripped of detail  to aid the development of a simplified model as well as to increase process understanding.

**1.2.4 Line 143**

**RC:** *So riming of droplets on cloud ice is neglected?*

AR: *Yes, the current CMP scheme does not contain riming of cloud droplets on cloud ice.*

**1.2.5 Line 146**

**RC:** *But snow number should not change at all for riming, of course. I assume that's the case for the parameterization here?*

AR: *This is correct, but the statement refers only to IC and CD concentrations. As for self-collection the IC mass is not changed, we have added a specification:*

> The implementation of these processes in terms of changes to the ice crystal and cloud droplet mass is detailed in Lohmann and Roeckner (1996), while the implementation of changes to the ice crystal and cloud droplet number concentration is simply in proportion to the mass changes (except for where the mass concentration is unaffected; Lohmann et al. (1999) and Lohmann (2002)).

**1.2.6 Line 147**

**RC:** *So cloud ice sedimentation is neglected completely? Later on lines 181-182 it seems that sedimentation \*is\* included, but reword earlier around line 147 to make this clear.*

AR: *We have specified how ice crystals sediment accordingly:*

> *Snowflakes precipitate, while ice crystals are smaller and  sediment but do not survive outside of clouds.*

**1.2.7 Line 158**

**RC:** *Assumed to be \*all\* liquid at mixed-phase temperatures? Also, better to specifically state the temperature range rather than saying "mixed-phase temperatures" which is imprecise.*

AR: *We have specified this accordingly:*

> *The detrained cloud particles are now assumed to be* all *liquid at mixed-phase temperatures* $(0\,°\mathrm{C} < T < -35\,°\mathrm{C};$ Dietlicher et al. (2019) and Muench and Lohmann (2020)).

**1.2.8 Line 164**

**RC:** *15 micron volumetric radius seems very low for a maximum-allowed value. Is this a typo or is this actually the value used? I understand this is only for determining the minimum CDNC, but with such a small maximum radius this implies a rather large CDNC. I realize this isn't a central part of the study but it seems worth pointing out.*

AR: *In the model a minimum CDNC value is needed likely because some aerosol sources are missing. As detailed in the manuscript we have replaced a fixed minimum CDNC value with a fixed maximum cloud droplet radius. $15\,\mathrm{\mu m}$ is the actual value used for the cloud droplet radius. Admittedly, cloud droplets larger than that can be found in the atmosphere. However, the parameter is set not to match the highest realistic value but to tune towards satisfactory model performance. From this perspective, a maximum cloud droplet radius of $15\,\mathrm{\mu m}$ provides a reasonable grid cell mean value.*

**1.2.9 Lines 184-186**

**RC:** *I think this sentence is confusing: "the gain of ice crystal concentrations in the level into which the ice crystals sediment is restricted to the closest loss of in-cloud ice crystal mass and number concentration in the levels above." Can this be clarified, perhaps using a more mathematical description which would be less confusing?*

AR: *We have augmented the sentence with a mathematical description:*

> *While the underlying problem of a weak sublimation needs to be addressed with future efforts, we introduced a correction of the sedimentation routine: the gain of ice crystal concentrations in the level $i$ into which the ice crystals sediment, $\Delta\mathrm{ICNC}_{\mathrm{sed},i}$ is restricted to the*  *loss of in-cloud ice crystal*  *number concentration in the*  lowest model level above level $i$ that lost ice crystals by sedimentation:*
>
> $$\Delta\mathrm{ICNC}_{\mathrm{sed},i} \leq -\Delta\mathrm{ICNC}_{\mathrm{sed},j}, \max\left(j \,|\, \Delta\mathrm{ICNC}_{\mathrm{sed}} < 0, j < i\right) \tag{1}$$

**1.2.10  Line 204**

**RC:** *I don't follow the logic that if the response to $\eta_i$ follows a sigmoidal curve, then the process need be only represented roughly. Do you mean because the relationship is monotonic? A sigmoidal curve of course could still have a sharp increase with small changes in $\eta_i$ implying a strong sensitivity to the given process. Or am I missing something here?*

AR: *This is a terminological mistake, we meant a curve where the slope around $\eta = 1$ is 0, i.e. a critical point. We have corrected this in the manuscript and added a sketch to illustrate the point (Fig. 2 in the new manuscript).*

> For example, if the model output variable (e.g. ice water path, IWP) as a function of $\eta_i$ has a critical point at $\eta_i = 1$ (i.e. slope of zero), this suggests that the process $i$ needs to be represented only  approximately and that some detail could probably be removed from its parameterisation without much of an effect on the model performance.

**1.2.11  Figure 2**

**RC:** *Perhaps mention PPE in the figure itself, not just the caption. i.e., the diagonal arrow could be labeled "Run ECHAM-HAM top generate the PPE" or something like that. Just a suggestion, the authors can take it or leave it.*

AR: *We welcome this suggestion and have added the term as suggested.*

**1.2.12  Line 235**

**RC:** *I assume "precipitation" here is surface precipitation?*

AR: *Your assumption is correct and we have specified this accordingly.*

> From the PPE, we can construct a surrogate model for every output variable that we are interested in by training a separate emulator for each output variable (ice crystal and cloud droplet number concentration, ice and liquid water path, shortwave and longwave cloud radiative effect, cloud cover, surface precipitation, ice, liquid and mixed-phase cloud cover).

**1.2.13  Lines 258-259**

**RC:** *This seems like an important point, but could you be more specific what you mean by "disruptive changes" in the CMP processes compared to the changes in Johnson et al.(2015)? Do you mean you use a larger range, including shutting processes off entirely by setting $\eta_i = 0$? Or something else?*

AR: *We have included a specification on what we think causes more disruptive changes in our setup compared to that in Johnson et al. (2015):*

> *We attribute this to the disruptive changes that the CMP process  perturbations induce as compared e.g. to the aerosol and CMP parameter changes applied by Johnson et al. (2015) (which did not include ice crystal autoconversion and perturbed parameters only within uncertainty bounds instead of whole processes), as well as to the fact that the simulations were not nudged.*

*In the original manuscript, we were also thinking of the changes induced by $\eta_i = 0$ but this part of the parameter space is no longer included in the PPE in the new manuscript.*

**1.2.14  Line 304**

**RC:** *This result doesn't seem that surprising – by suppressing "aggregation" you're likely decreasing the snow mixing ratio substantially, in turn reducing riming since the amount of riming depends on the amount of snow. This result likely reflects the somewhat artificiality of neglecting riming of cloud ice (at least, I think it's neglected in the scheme here), and more generally the separation of ice into "ice crystal" or "cloud ice" and "snow" categories (major comment #4). This kind of "thresholding" behavior associated with "aggregation" is discussed a bit below on line 309, but this is unphysical and reflects an ad-hoc separation of ice into cloud ice and snow categories. See major comment #5 above.*

 AR: *We agree with your interpretation that aggregation inhibits riming because the latter depends on the snow that the former produces. As discussed above this is inherent to the CMP scheme.*

**1.2.15  Lines 309-310**

**RC:** *Snow is not generated by freezing of rain? In a climate model this will be tiny, but it should still be non-zero. Or else, how is rain formed/lofted above the freezing level treated?*

 AR: *No, in this CMP scheme rain reaches the surface within one timestep and thus cannot freeze to form snow.*

**1.2.16  Line 322**

**RC:** *I disagree with the sentence: "In classical sensitivity studies, where processes are only turned on and off, only the large signal induced by aggregation would have been visible." Clearly there would still be a large signal to turning off the aggregation process completely.*

 AR: *Indeed this is what the sentence says: in classical sensitivity studies (. . . ) the large signal induced by aggregation would have been visible.*

**RC:** *A few lines down this sentence also isn't clear: "Recognizing it as a threshold process and seeing the gradual response to small deviations from 1.0 in $\eta_{\mathrm{aggr}}$, it appears that there is potential for a less accurate description of aggregation in the model." To me, discussing a "threshold" process as a "gradual response" is self-contradictory. Not clear exactly what you mean by aggregation as a "threshold process".*

 AR: *What we mean is illustrated in the sketch we added (Fig. 2 in the new manuscript): no matter what the response is far away from $\eta = 1$, if the slope is close to 0 around $\eta = 1$ there is some potential for a less accurate description of the process in question (purple line in the sketch).*

**1.2.17 Lines 343-345**

**RC:** *I'm not sure if this comparison with Lohmann and Ferrachat (2010) is meaningful because here you're choosing an ad-hoc range to vary the $\eta_i$'s. And the range extends to unrealistic values, such as setting $\eta_i = 0$ meaning the process is shut off completely. In contrast, I assume Lohmann and Ferrachat (2010) varied scheme parameters over a range of physically plausible values?*

 **AR:** *Lohmann and Ferrachat (2010) varied tuning parameters over plausible tuning ranges. For two of the values, the perturbations span multiplicative ranges of 10 or larger. As the updated PPE design in our study includes perturbations between 0.5 and 2 (now excluding $\eta + i = 0$, a comparison to Lohmann and Ferrachat (2010) is plausible.*

**1.2.18 Line 346**

**RC:** *It's not clear what you mean by varying the autoconversion rate between 1 and 10. Autoconversion rate should have units. Do you mean you vary it by a factor from 1 to 10?*

 **AR:** *Indeed the autoconversion rate has been multiplied by a factor between 1 and 10 in Lohmann and Ferrachat (2010), and we have specified this accordingly:*

> Only for LWP Lohmann and Ferrachat (2010) find a larger range of about $50\,\mathrm{gm}^{-2}$ when they  multiply the autoconversion rate with a factor between 1 and 10. As this warm-rain process is not included in the present analysis, it is reasonable that the observed variation for LWP is smaller.

**1.2.19 Line 400**

**RC:** *I would suggest using a different variable than "f" for the data and complex coefficients. It's confusing to use the same symbol "f" for both.*

 **AR:** *To remain in line with common notation but avoid ambiguity we have changed the symbol for the function to $F$.*

> Mathematically, the model data can be represented as a linear combination of the orthogonal spherical harmonics basis functions as follows:
>
> $$f(\theta,\phi) = \sum_{l=0}^{\infty} \sum_{m=-l}^{l} \underline{f}F_l^m Y_l^m(\theta,\phi) \qquad (2)$$
>
> The data $f$ is then a function of the longitude $\theta$ and latitude $\phi$, with $Y_l^m$ a spherical harmonics function of degree $l$ and order $m$ ($l$ and $m$ are integers, with $-l \leq m \leq l$). The complex coefficients $\cancel{f_l^m}\underline{F_l^m}$ can be computed as:
>
> $$\underline{f}F_l^m = \int_\Omega f(\theta,\phi) Y_l^m(\theta,\phi)\mathrm{d}\Omega \qquad (3)$$

**1.2.20 Line 465**

**RC:** *Not sure I'd say you're multiplying the process "effects" by a factor, but rather you're multiplying the process rates.*

**AR:** *We have changed the formulation accordingly to make this more clear.*

> Different from previous studies (e.g. Wellmann et al. (2020) and Hawker et al. (2021)) we  perturb the four CMP processes of autoconversion, riming, accretion and self-collection of ice as a whole. This is achieved by multiplying their  process rates with a factor between  0.5 and 2.

**1.3. Editorial comments**

- Lines 236-237: Would this sentence be better as: "For the kernel, an additive combination of the linear, polynomial, bias and exponential kernels was used (Duvenaud, 2014)."?

- Line 240: I think "perturb" should be "perturbed", past tense.

- Line 270: I think you can remove "thus".

- Line 356: "verion" is misspelled.

- Line 396: I think there should be a comma after "decomposition".

- Line 513: It seems "process" should be "processes".

**AR:** *Thank you for pointing out these errors. We corrected them in the new version of the manuscript.*

**2. Reviewer Comment #2**

**RC:** *This study makes use of the technique of variance-based sensitivity analysis on an emulated perturbed parameter ensemble of the global aerosol-climate model ECHAM-HAM to understand the impact of perturbing the effect of selected cloud-ice microphysical processes on the models' output. For each process considered, a 'phasing parameter' is implemented to perturb the strength of the generated effect of a process (from 0 to 200%), and the study uses this 'phasing' as a proxy for the effect of process simplification. This is a novel use of the emulation and sensitivity analysis approach to assess model behaviour under uncertainty. The paper definitely falls within the scope of ACP and EGU, and is written to a high standard. However, there are several points that I believe need clarification (see specific comments below). In particular, I am concerned that the design and sampling for the PPE simulations does not provide the required coverage of the actual phasing effect for a completely robust analysis. The parameters are multiplicative factors but they are not treated as such in the PPE design, so the PPE has a very skewed coverage over the effect of 'phasing out' a process (see specific comment at Line 210-214, below). Because of this, the PPE looks to have very low coverage of training data where the 'phasing out effect' is strongest and the model response is likely to be greatest/more erratic (as the $\eta_i$ parameters move towards zero), and a much denser coverage of training points where the phasing out effect is weaker ($0.5 < \eta_i < 1$) or there is over-estimation of a process due to an inaccurate description ($\eta_i > 1$). Given the low amount of training*

*information for the emulator where the phasing out is strong, I'm not convinced that the emulator can properly capture this response in any kind of detail. For a more robust conclusion, I would recommend (if possible - this would be a major revision) a re-design of the PPE input combinations to properly cover the parameter space for emulation and provide a more even sampling of the 'phasing out effect' for the sensitivity analysis. Once this and the further issues/comments below are addressed, I would recommend the publication of the manuscript in ACP.*

AR: *Thank you for your thoughtful and well explained feedback. Please find the answer to your comments below.*

**2.1. Specific Comments**

**2.1.1 Line 7 (in abstract)**

RC: *'The response to the phasing of a process thereby serves as a proxy for the effect of a simplification'. This sentence is confusing me in two ways. Firstly, what is meant by 'the phasing of a process'? – this is unclear (I realise this might be explained in the paper, but people will read the abstract first, so it's not clear at this point.). Also, the use of the word 'thereby' in this sentence is confusing – it suggests that the information in this sentence follows as a result of the sentence before it, but I don't think it is – it is a separate point with new information about the method/assumptions made. Please remove 'thereby' and re-phrase to clarify.*

AR: *In response to Reviewer #1 we have exchanged the "phasing" for "perturbation" and we have removed the "thereby" as you suggested:*

> The response to the  perturbation of a process  serves as a proxy for the effect of a simplification.

**2.1.2 Line 13 (in abstract)**

RC: *Is this really a 'new framework'? Statistical emulation and sensitivity analysis have been used in several studies to assess process impacts in complex models of clouds, the atmosphere and the climate, as you have stated in the paper e.g. Line 239-240: 'This approach is similar to Johnson et al. (2015)...,' Please re-phrase.*

AR: *We have rephrased to a new application instead:*

> *This study introduces a new  application for the combination of statistical emulation and sensitivity analysis to evaluate the sensitivity of a complex numerical model  to a specific parameterized process. It paves the way for simplifications of CMP processes leading to more interpretable climate model results.*

**2.1.3 Line 42-44**

RC: *'Finally, the detail of ...increases computational demand and thereby costs or inhibits other advancements such as the move towards higher resolution...'. Do parameterisations at a lower resolution still*

*always hold at a higher resolution? Is it not the case that for a move to higher resolution, we need more detailed representations of processes? – So, to a reasonable extent, doesn't higher resolution and more detailed process descriptions have a dependence? – I don't think they are quite as independent as this sentence is suggesting – please clarify.*

AR: *Since the CMP parameterizations have the treatment of very small particles at their core (Morrison et al., 2020), many of the formulations can stay at higher resolution. However, some assumptions, e.g. that precipitation reaches the surface within one model timestep, break down when moving to higher resolution. Also the tuning of the model will need to change when moving to higher resolution (Pincus and Klein, 2000). We have adapted the sentence you point to in order to reflect that interdependence:*

> *Finally, the detail of the aerosol and cloud microphysics increases computational demand and thereby costs  (though anticipating the results of Sec. 3.6, the four CMP processes investigated in this study require negligible computing time). It can thereby inhibit other advancements such as the move towards high-resolution simulations (which may themselves also require adaptations of the CMP schemes) or larger ensembles.*

**2.1.4  Line 63**

RC: *'. . . a simplified model equifinal to a more complex model'. What does 'equifinal to' mean here, when the difference is between 2 different models (simplified and complex)? – I'm not sure if this is the same as the definition for equifinality on page 2, where different parts of a model's parameter space lead to a similar observed state? Please clarify.*

AR: *We have added such a specification in the text:*

> *However, due to the reasons mentioned above, a simplified model equifinal to a more complex model may be more useful for gaining understanding of climate models (equifinal meaning that the two model versions lead to similar results).*

**2.1.5  Line 68**

RC: *'. . . The influence of CMPs has been shown to dominate over that of aerosol schemes. . . ' I'm not sure this has always been the case for a GCM? For example, Regayre at al (2018) [Figure 9] showed that both aerosol and physical atmosphere (cloud-related) model parameters are both important sources of uncertainty in aerosol ERF in the GCM HadGEM3-UKCA. Please update the text here to reflect this.*

AR: *We have adapted the text to include this finding:*

> *The role of CMPs within GCMs has been investigated previously: The influence of CMPs has been shown to dominate over that of aerosol schemes in affecting clouds and precipitation in the Weather Research and Forecast model (White et al., 2017), as well as to dampen the influence of aerosol microphysics on cloud condensation nuclei and ice nucleating particles  in a regional model (Glassmeier et al., 2017). For the HadGEM-UKCA global aerosol-climate model, Regayre et al. (2018) have shown that both aerosol and physical atmosphere parameters contribute to uncertainty in aerosol effective radiative forcing.*

**2.1.6   Line 89**

**RC:** *'... variations in input as well as...': The word 'input' should be plural. Also, should this be '... variations in independent inputs...'? Most global sensitivity analysis techniques (especially variance-based sensitivity analysis) assume independence between inputs.*

 **AR:** *This is correct and we have corrected it in the text as follows:*

> *It allows to divide the total variation in output into the direct contributions from variations in  independent inputs as well as  from their interactions.*

**2.1.7   Line 112**

**RC:** *'By phasing we mean that we vary the effectiveness of a given process, going from using 0 to 200% of a process's effect in the model'. This is quite a difficult concept to understand here in terms of how this can be done – I don't think all processes within a model could be easily 'phased'. What is meant by 'effectiveness'? How is it defined and is this 'effectiveness' the same or does it differ between inputs / processes? I realise that the next section (2) will bring more detail on this, but giving a small (brief) example or a little more detail here could provide a bit more clarity for the reader as a starting point.*

 **AR:** *We have added an explanatory example. In combination with our response to Reviewer #1's comments (replacing phasing with perturbing) we have adjusted the paragraph in question as follows:*

> *By  perturbing we mean that we vary the effectiveness of a given process, going from using  50 to 200% of a process's effect in the model. For example, if a process affects the ice crystal number concentration, the change induced on it is multiplied by a perturbation factor between 0.5 and 2 in each timestep. This means that in the extreme cases it would produce half or twice the effect on the ice crystal number concentration that it has in the default model (see Sec. 2.2 for further detail).*

**2.1.8   Line 202 (which also connects the point for Line 112)**

**RC:** *'From the response of model output to variations in $\eta_i$, we can extract how accurately a process i needs to be represented in the model.' How can you extract this? Does process accuracy actually directly correspond to the effect of 'phasing' in/out a process like this? From the abstract: 'The response to*

*the phasing of a process serves as a proxy for the effect of simplification' – But, is a less accurate / simplified process necessarily going to produce a reduced change in the additional 'delta' component in equation 1? Couldn't a simplified process potentially make that component larger? Or, have any effect on what that value is? I cannot work out if it really is realistic to treat a process in this way. Please give a clearer description as to how/why the phasing feeds through to inference on process accuracy and process simplification.*

AR:    *As you suggested below we have added a sketch to illustrate our point (new Fig. 2). Process accuracy does not correspond to the perturbation directly. You are right that a simplified process could reduce or enlargen the delta component, and this change would likely be variable in space and time. The perturbations we apply generate results, and from the shape of these results and the sensitivity analysis we can deduce how sensitive the model is to any changes in the process in question. This shape of the results/sensitivity is the proxy for what would happen if one simplified or approximated the process.*

**2.1.9    Line204 (connects to the point for Line 202)**

RC:    *What is a 'sigmoidal function'? [Will a general reader know?] From google, a 'sigmoid function' has a loose 'S' shape? [like the 'logistic function: f(x) = 1/(1+exp(-x))]. So, it's gradient can be steep or shallow depending where you are on the curve? Hence, how can you know that some detail can easily be left out? This needs more clarity – how this parameter for each process can inform the need for model complexity is a key message from the study, so understanding how to interpret it is very important, yet it seems to be skipped over here. A diagram to help the reader picture what you mean (maybe with several different options as to how the parameter $\eta_i$ could be interpreted for a process) would be helpful, as it is not clear to me that the statement in this example (lines 203-205) is true, or how the parameter in general will inform us.*

AR:    *As Reviewer #1 also pointed out, what we meant is not a sigmoidal function but a curve where the slope around $\eta = 1$ is 0, i.e. a critical point. We have corrected this in the manuscript and added a sketch to illustrate the point (Fig. 2 in the new manuscript).*

**2.1.10    Lines 210-214**

RC:    *The scaling of the '$\eta$' parameters here treats them as linear factors, but I don't think they are. I think each $\eta_i$ is a multiplicative factor, and as such, the phasing effect is not varying evenly over the $\eta$ ranges, with it likely that there will be a much more significant effect on the model behaviour with very small values as an $\eta$ approaches 0. [I think this is also an aspect of the cause of the large outlier at the very low $\eta_{\mathrm{aggr}}$ in Fig 3?]. This is tricky to explain, but within your range of 0<$\eta$<2, 0.5<$\eta_i$<1 corresponds to a scaling of the given process by 1 times (1x) to a half times (0.5x), covering a 'phasing reduction' of the process by up to 2 times (2x) smaller than its default effect over a range in $\eta$ values of 0.5. But, lower down the $\eta$ range, say 0.01<$\eta$<0.1, this covers a more significant reduction of 10 times smaller (0.1x) to 100 times smaller (0.01x) than the default effect, but within a much smaller range on $\eta$ of size 0.09. Because of this, your PPE looks to have very low coverage of training data where the 'phasing out effect' is strongest and the model response is likely to be greatest/more erratic (as the $\eta$ parameters move towards zero), and a much denser coverage of training points where the phasing out effect is relatively weaker (0.5<$\eta_i$<1) or where you consider over-estimation (1<$\eta_i$<2). In fact, designing the training points linearly between 0 and 2 leads to having approx. 50% of simulations with $\eta$>1 for each $\eta$ parameter – so really concentrating on the parts of the ranges / 4-d parameter space that is to 'imitate an overestimation*

*of a given process due to an inaccurate description' (Line 214). Is that what you intended? As, my understanding is that this area of the space isn't really the focus of the study (to understand how sensitive the model responses are to phasing out processes), so why sample it the most? I think this is a significant error in your PPE design. And this will also feed through to affect how you sample the phasing effects for the sensitivity analysis (concentrated away from a strong phasing out, and highly focussed on $\eta_i>1$, if sampling uniformly). In most PPE studies, parameters like this are varied on a log 10 scale to account for the multiplicative behaviour. However, including zero in your range means a log 10 transform is difficult here (as log 10 (0) = -inf) – it might be better to only vary the parameters down to a small value close to zero (e.g. 0.001) so that a log 10 scaling could be used to even out the phasing effect over the $\eta$ ranges. Given the low amount of training information for the emulator where the phasing out is strong, I'm not convinced that the emulator can properly capture this response in any kind of detail. If possible, for a more robust conclusion, I would recommend a re-design of the PPE input combinations in this way to properly cover the parameter space for emulation, provide a more even sampling of the 'phasing out effect' for the sensitivity analysis, and also provide more detail on how the model response changes as a process is phased out. If this is not possible, please at least acknowledge the assumption that has been made here – that you treat the $\eta$ parameters as linearly varying factors – and note/describe/discuss here and in the results and discussion section how this is affecting your analysis and results.*

AR: *Thank you for pointing us to this error. We have revised the PPE design to sample logarithmically from 0.5 to 2 to address your concern. As you correctly point out our focus is on the region around $\eta = 1$, which is why we now constrained the parameter space to a multiplicatively equal range of 2 on either side of $\eta = 1$. The extreme case of $\eta = 0$ is still covered with the one-at-a-time sensitivity tests displayed in e.g. Fig. 4 but no longer included in the PPE to avoid overinterpretation. Much of the analysis and results proved to be robust to this change. However, for the spatial analysis the emulators default for more variables. This is reasonable since the PPE now covers a smaller region of the phase space with a less intense signal.*

**2.1.11    Line 215**

RC: **The phrase 'sets of simulation input' is unclear. I think you mean 'the set of input parameter combinations ($\eta_1$, $\eta_2$, $\eta_3$, $\eta_4$) to be simulated with the model'. Please clarify the text.**

AR: *To clarify, we have used your suggested formulation:*

> *To probe the multi-dimensional input parameter space effectively, the sets of  input parameter combinations ($\eta_1$, $\eta_2$, $\eta_3$, $\eta_4$) to be simulated with the model were generated with Latin Hypercube Sampling (LHS, using the Python library PyDOE (tisimst, 2021)), which maximizes the spacing between inputs and provides good coverage of the parameter space, even when only a few input parameters are important (Morris and Mitchell, 1995).*

**2.1.12    Line 236**

RC: **'As kernel, an additive combination of the linear, polynomial, bias and exponential kernel was used (Duvenaud, 2014)' What does 'as kernel' mean? Is this the function that describes the covariance between points in the Gaussian process (GP), and so control the smoothness of the GP response surface? This additive combination seems rather complex – why is this chosen/used?**

AR: *Exactly, the kernel is the covariance function that serves as a prior for the functions that the GP can represent (Watson-Parris et al., 2021b). The combination was selected in preliminary tests as the one giving the best validation in emulating the response surface. We have added specifications to this in the new manuscript:*

> * For the kernel (or covariance function, Watson-Parris et al., 2021b), an additive combination of the linear, polynomial, bias and exponential kernels was used as this performed best in preliminary tests (not shown, Duvenaud (2014)).*

**2.1.13  Line 237**

RC: *'The input data was centred and whitened prior to emulation'. What exactly does that mean? Why is this needed, and how does it affect the emulator / surrogate model? Please give more detail. [There isn't enough detail here for someone to be able to replicate the analysis.]*

AR: *We have added more detail and removed the unnecessary technical terminology as follows:*

> *Other model specifics were set as default in  Watson-Parris et al. (2021a). As the emulation operates best on standardized data with zero mean and unity variance, the mean was removed from the input data, which was then scaled by dividing it by the standard deviation, prior to emulation.*

AR: *In addition, we hope that the scripts that we supply on the open access data repository zenodo will help anyone to replicate the analysis.*

**2.1.14  Line 245**

RC: *'1-out validation'. This is an unusual term to describe this approach. Please change to 'Leave-one-out validation', here and elsewhere.*

AR: *Thank you for alerting us to this terminology. We have exchanged it accordingly.*

**2.1.15  Line 246-247**

RC: *In the brackets, please use the notation as it is in the formula. So, '(with $Y_{\mathrm{sim}}$ and $Y_{\mathrm{emu}}$ the output of the ECHAM-HAM simulations and the emulated output respectively, and $V_{\mathrm{emu}}$ the emulator variance)'*

AR: *We have adapted the text accordingly:*

> *In Fig. 4 a) and b), the individual standardized errors, $\frac{Y_{\mathrm{sim}} - Y_{\mathrm{emu}}}{\sqrt{V_{\mathrm{emu}}}}$ (with $Y_{\mathrm{sim}}$ and $Y_{\mathrm{emu}}$ the output of the ECHAM-HAM simulations and the emulated output, respectively, and $V_{\mathrm{emu}}$ the emulator variance), are plotted against the emulated output and input parameters.*

**2.1.16 Lines 257-266**

**RC:** *I think this could also result in part from the PPE design and low coverage of large changes in the phasing amount at the low end of the parameter ranges (see comment [L210-214] above), which should also be acknowledged here.*

AR: *As we have improved the PPE design based on your comments above, this reasoning no longer applies here.*

**2.1.17 Line 299 (and in the paragraphs that follow)**

**RC:** *'...inflicted by the inhibition of the other three processes.' I don't think 'inhibition' is the right word to use here – what do you mean by a process' 'inhibition'? Do you mean it has very little effect? Or just the process of 'phasing out'? – it's not clear. [When I google it, I don't find a relevant meaning for this context.] Please re-phrase and remove this term 'inhibition' throughout the manuscript.*

AR: *We reworded the sentence since what we meant was simply that $\eta_{\mathrm{aggr}} = 0$ has a larger effect than the other $\eta_i = 0$:*

> *Of the four  perturbed processes, turning off  autoconversion has the largest effect on model output: the global annual mean ice water path (IWP) is more than doubled, and the increase in cloud cover and decrease in precipitation dwarf the changes inflicted by  turning off the other three processes.*

**2.1.18 Line 305**

**RC:** *'The shape of the model response to the gradual phasing of the processes holds additional information: while the generated model response is mostly gradual, for low $\eta_{\mathrm{aggr}}$ the response is more abrupt.' – This is also, in part, the effect of the uneven distribution of the 'phasing effect' over the parameter range (see comment [L210-214] above), which should be acknowledged here.*

AR: *As we have improved the PPE design based on your comments above, this reasoning no longer applies here.*

**2.1.19 Line 310**

**RC:** *'As can be seen from Fig. 1 it (aggregation) is the only process that generates snow flakes. Accretion and riming need the snow flakes to be able to act upon them.' Does this mean that there is a dependence between the phasing parameters here? Is this a strong dependence? i.e. without quite a high value of $\eta_{\mathrm{aggr}}$, you cannot have (it isn't realistic to have) a high value of $\eta_{\mathrm{accr}}$ or $\eta_{\mathrm{rim}}$? – or can you have (is it realistic to have) a high value of $\eta_{\mathrm{rim}}$ when $\eta_{\mathrm{aggr}}$ is pretty small (just not zero)? If it is a strong dependence, then this would invalidate the assumptions of the variance-based sensitivity analysis (Sections 2.5,3.3) which assumes independence between inputs for the breakdown of the variance into its component parts.*

AR: *The four parameters that we vary are independent: no matter what the modulation of the aggregation effect, we can modulate the riming effect completely independently from that. Only when aggregation is turned off, varying $\eta_{\mathrm{rime}}$ does not have any effect, but this is now excluded from the PPE design.*

**2.1.20 Figure 6 / Figure 7 captions**

**RC:** *It isn't correct to label the panels in these figures as 'correlation matrix'. They are not plots of correlations – they are 2-d projections of the sampling of the 4-d response surface for a given output. Please amend the captions.*

AR: *We have implemented an adaptation of your suggestion as the shown plots are three dimensional and the whole response surface is five dimensional:*

>  Three dimensional projections of the  response surface of the emulated PPE.

**2.1.21 Line 353-355**

**RC:** *'The observed sensitivities are different from what Bacer et al. (2021)find.... In our analysis, the influence of aggregation dwarfs that of accretion in terms of sensitivity indices as well as for the process rates...' Are you comparing 'like-for-like' here? Or are you seeing a larger effect for aggregation because you vary the process more – to phasing it out completely? Please clarify.*

AR: *You are right that the sensitivity indices are not directly comparable to the findings in Bacer et al. (2021) as the former are derived from process perturbations. However, the dark blue columns in our Fig. 5 are diagnosed for the default simulation, which is the same setup as used for Bacer et al. (2021). We have specified this in the text as follows:*

>  The sensitivity indices are not directly comparable to Bacer et al. (2021). However, for the default simulation the process rates are diagnosed as in Bacer et al. (2021) and thus comparable.

**2.1.22 Line 363-364**

**RC:** *'This was excluded from the sensitivity analysis as only the input parameter space with $\eta_{\mathrm{aggr}} \geq 0.5$ was taken into consideration...' Is it not more appropriate to consider the sensitivity analysis (SA) for a range of $\eta_i$ that doesn't go all the way down to zero anyway? Is it not the case that processes need to still be accounted for (they still need to be included in the model), but that you are investigating just how detailed or not (phased in or out) that representation needs to be? Why did you choose 0.5 here? Also, is the focus of the SA in terms of space sampled more on $\eta_i > 1$? Could this be biasing the SA results away from the effect that you really want to consider? (i.e. is it focussing much less on phasing out from the current full complexity at $\eta=1$, and more on the effects of increasing complexity / overestimation of a process?). How might this feed through to affect the inferences and conclusions made?*

AR: *We have addressed this point with a redesign of the PPE. We now vary the perturbation parameters $\eta_i$ logarithmically between 0.5 and 2 so as to sample the "phasing in" and "phasing out" part of the parameter space equally. The cases with $\eta_i = 0$ are included only in the one-at-a-time sensitivity tests.*

**2.1.23 Line 388**

**RC:** *should this say '... representation of global annual mean IWP and LWP, the....'?*

AR: *In response to comment # 1.1.2 this sentence has changed substantially. To answer your question: yes, at this point of the analysis one could only derive statements on the global annual mean values.*

**2.1.24 Line 392**

**RC:** *'... analysis of grid-point level data is tedious and error prone...' I agree that you have to create a lot of emulators to do grid-point analysis, so reducing the dimensions as you do in this section is definitely advantageous. But, why is the grid-point analysis 'error-prone'? Given the quite high level of uncertainty in emulator prediction from the emulators of the spherical harmonics here (Line 429 and Appendix D Figure D1 part d, where the points are rather widely scattered about the line of equality), is it really fair to infer that this approach is less error-prone? Also, is it possible to generate a plot (map) of how the sensitivities of parameters vary over the globe via the method used here, like you might with a grid-point level analysis? (I'm not suggesting you necessarily do this here – I'm just wondering if it is possible to do this...)*

AR: *The emulation and subsequent sensitivity analysis of grid-point level data is more error-prone, because the emulation method from Watson-Parris et al. (2021b) does not take spatial correlation into account. As each grid point is evaluated only by itself, there is more noise in the data that is reduced by reducing the dimensions of the data set. Therefore, seeing that the validation of the spherical harmonics is already difficult because of a small signal, we assume that this problem would be elevated at the grid point scale. We have specified this in the new manuscript as follows:*

> *Since the emulation and subsequent analysis of grid-point level data is tedious and error-prone due to the small signal and large noise, we compress the information in the data to a space of lower dimensionality.*

AR: *No, a map of how the sensitivities of the parameters vary over the globe is not possible with this method. This is because we apply the sensitivity analysis onto the angular amplitude spectrum, which is a combination of all $m$ for each $l$, and thus we cannot do the inverse transformation.*

**2.1.25 Line 409**

**RC:** *Should the sum term here have '(l)' at the end, as it does on line 406?*

AR: *You are right, we have added the "(l)" as you suggested.*

**2.1.26 Line 427 (end)**

**RC:** *should this say 'LWP and CDNC are dominated by....'?*

AR: *You are right. However, with the new PPE design the small-scale emulation of CDNC does not validate, so the sentence was adjusted to the following:*

> *The  LWP is dominated by riming on all regional scales and on the global scale, while  at some degrees $l < 7$  it is also heavily influenced by autoconversion.*

**2.1.27 Line 443**

**RC:** *'...as with the 48th member the computational constraint was too tight for the emulator' I don't understand what you mean by 'computational constraint was too tight'. Please clarify.*

AR: *In terms of practical understanding, what we mean to describe is the fact that for some of the emulations, the GP was only able to fit to 47 members of the PPE. Adding a 48th member led to an overdescribed system, which in practical terms manifested in a non-invertible matrix error in the code. We have tried but failed to eliminate this error by adjusting the GP model properties, and understanding it further is beyond the scope of this paper as it is embedded in libraries underlying the used software.*

**2.1.28 Figures 10 and 11**

**RC:** *Why do you only show the total sensitivity index? Isn't the First order effect more informative? With the total sensitivity index, if there are large interactions, or if the model output is quite noisy which can sometimes induce interaction in the indices that is not real, this index on its own can give a potentially skewed/false impression of the true sensitivity. To conclude robustly on the sensitivity, really you should present both indices. Can the first order index also be shown in Figures 10 and 11?*

AR: *Because the first order sensitivity index looked very similar to the total one, we did not include it for clarity/brevity of the Figures. For added transparency we have now included the first order sensitivity index plots in the Appendix.*

**2.2. Technical Corrections**

- Line 9 (in abstract): For clarity, change '...on snow influences mostly the liquid phase.' to '...on snow **has most influence** on the liquid phase.'

- Line 30: Change 'More complexity has also its downsides: ...' to 'More complexity **also has** its downsides: ...'

- Line 65: Change '...in face of...' to '...in **the** face of...'

- Line 136: Missing bracket at the end of the sentence: '...see Neubauer et al. (2019)).'

- Line 186: '...in-cloud ICNC concentration and snow...' Remove the work 'concentration' here, as it is already in the acronym ICNC.

- Line190: Change '...Table A1 in the Appendix.' to '...Table A1 in Appendix A.'

- Line 227: 'Each simulation included a 3 months spin up...'. Change to either 'Each simulation included 3 months **of** spin up...' or 'Each simulation included a 3 month spin up...'

- Line 230: 'Fig. 1e' should be 'Fig. **2**e'. Please correct.

- Line 256: Change '. . . bounds of the emulator crossing. . . ' to '. . . bounds o**n** the emulator **predictions** crossing. . . '

- Figure 3 caption, line 1: Remove 'according to Bastos and O'Hagan (2009).' from the first sentence. (Bastos and O'Hagan do not validate this emulator. . . )

- Line 349: Missing word: '. . . allows **us** to quantify. . . '.

- Line 362: Missing word: '. . . space and not due **to** the threshold behaviour . . . '

- Line 382: Remove the word 'as' before 'e.g.'.

- End of figure 9 caption: Change 'truncated' to 'truncate'.

- End of figure 10 caption: Change 'is missing here' to 'are missing here'.

- Line 608: Check the details of the reference 'Hawker et al (2021a)' – This paper has now been accepted in ACP and should be published soon, so the exact reference might be available?

AR:   *Thank you for these corrections. We have incorporated them as suggested.*

**References**

Archer-Nicholls, S., N. L. Abraham, Y. M. Shin, J. Weber, M. R. Russo, and ... (2021). "The Common Representative Intermediates Mechanism Version 2 in the United Kingdom Chemistry and Aerosols Model". In: *Journal of Advances in Modeling Earth Systems* 13.5, p. 50.

Bacer, Sara, Sylvia C. Sullivan, Odran Sourdeval, Holger Tost, Jos Lelieveld, and Andrea Pozzer (2021). "Cold Cloud Microphysical Process Rates in a Global Chemistry–Climate Model". In: *Atmospheric Chemistry and Physics* 21.3, pp. 1485–1505. ISSN: 1680-7324. DOI: 10.5194/acp-21-1485-2021.

Carslaw, Kenneth, Lindsay Lee, Leighton Regayre, and Jill Johnson (2018). "Climate Models Are Uncertain, but We Can Do Something About It". In: *Eos* 99. ISSN: 2324-9250. DOI: 10.1029/2018EO093757.

Dagon, Katherine, Benjamin M. Sanderson, Rosie A. Fisher, and David M. Lawrence (2020). "A Machine Learning Approach to Emulation and Biophysical Parameter Estimation with the Community Land Model, Version 5". In: *Advances in Statistical Climatology, Meteorology and Oceanography* 6.2, pp. 223–244. ISSN: 2364-3587. DOI: 10.5194/ascmo-6-223-2020.

Dietlicher, Remo, David Neubauer, and Ulrike Lohmann (2018). "Prognostic Parameterization of Cloud Ice with a Single Category in the Aerosol-Climate Model ECHAM(v6.3.0)-HAM(v2.3)". In: *Geoscientific Model Development* 11.4, pp. 1557–1576. ISSN: 1991-9603. DOI: 10.5194/gmd-11-1557-2018.

— (2019). "Elucidating Ice Formation Pathways in the Aerosol–Climate Model ECHAM6-HAM2". In: *Atmospheric Chemistry and Physics* 19.14, pp. 9061–9080. ISSN: 1680-7324. DOI: 10.5194/acp-19-9061-2019.

Duvenaud, David Kristjanson (2014). "Automatic Model Construction with Gaussian Processes". PhD thesis. University of Cambridge.

Eidhammer, Trude, Hugh Morrison, David Mitchell, Andrew Gettelman, and Ehsan Erfani (2017). "Improvements in Global Climate Model Microphysics Using a Consistent Representation of Ice Particle Properties". In: *Journal of Climate* 30.2, pp. 609–629. ISSN: 0894-8755, 1520-0442. DOI: 10.1175/JCLI-D-16-0050.1.

Fisher, Rosie A. and Charles D. Koven (2020). "Perspectives on the Future of Land Surface Models and the Challenges of Representing Complex Terrestrial Systems". In: *Journal of Advances in Modeling Earth Systems* 12.4. ISSN: 1942-2466, 1942-2466. DOI: 10.1029/2018MS001453.

Ghan, Steven J. et al. (2013). "A Simple Model of Global Aerosol Indirect Effects". In: *Journal of Geophysical Research: Atmospheres* 118.12, pp. 6688–6707. ISSN: 2169897X. DOI: 10.1002/jgrd.50567.

Glassmeier, Franziska, Anna Possner, Bernhard Vogel, Heike Vogel, and Ulrike Lohmann (2017). "A Comparison of Two Chemistry and Aerosol Schemes on the Regional Scale and the Resulting Impact on Radiative Properties and Liquid- and Ice-Phase Aerosol–Cloud Interactions". In: *Atmospheric Chemistry and Physics* 17.14, pp. 8651–8680. ISSN: 1680-7324. DOI: 10.5194/acp-17-8651-2017.

Hawker, Rachel E. et al. (2021). "The Temperature Dependence of Ice-Nucleating Particle Concentrations Affects the Radiative Properties of Tropical Convective Cloud Systems". In: *Atmospheric Chemistry and Physics* 21.7, pp. 5439–5461. ISSN: 1680-7324. DOI: 10.5194/acp-21-5439-2021.

He, Fei and Derek J. Posselt (2015). "Impact of Parameterized Physical Processes on Simulated Tropical Cyclone Characteristics in the Community Atmosphere Model". In: *Journal of Climate* 28.24, pp. 9857–9872. ISSN: 0894-8755, 1520-0442. DOI: 10.1175/JCLI-D-15-0255.1.

Johnson, J. S., Z. Cui, L. A. Lee, J. P. Gosling, A. M. Blyth, and K. S. Carslaw (2015). "Evaluating Uncertainty in Convective Cloud Microphysics Using Statistical Emulation". In: *Journal of Advances in Modeling Earth Systems* 7.1, pp. 162–187. ISSN: 19422466. DOI: 10.1002/2014MS000383.

Knutti, Reto and Jan Sedláček (2013). "Robustness and Uncertainties in the New CMIP5 Climate Model Projections". In: *Nature Climate Change* 3.4, pp. 369–373. ISSN: 1758-678X, 1758-6798. DOI: 10.1038/nclimate1716.

Lee, L. A., K. S. Carslaw, K. J. Pringle, G. W. Mann, and D. V. Spracklen (2011). "Emulation of a Complex Global Aerosol Model to Quantify Sensitivity to Uncertain Parameters". In: *Atmospheric Chemistry and Physics* 11.23, pp. 12253–12273. ISSN: 1680-7324. DOI: 10.5194/acp-11-12253-2011.

Lee, L. A., K. S. Carslaw, K. J. Pringle, and G. W. Mann (2012). "Mapping the Uncertainty in Global CCN Using Emulation". In: *Atmospheric Chemistry and Physics* 12.20, pp. 9739–9751. ISSN: 1680-7324. DOI: 10.5194/acp-12-9739-2012.

Lee, Lindsay A., Carly L. Reddington, and Kenneth S. Carslaw (2016). "On the Relationship between Aerosol Model Uncertainty and Radiative Forcing Uncertainty". In: *Proceedings of the National Academy of Sciences* 113.21, pp. 5820–5827. ISSN: 0027-8424, 1091-6490. DOI: 10.1073/pnas.1507050113.

Lohmann, U. and S. Ferrachat (2010). "Impact of Parametric Uncertainties on the Present-Day Climate and on the Anthropogenic Aerosol Effect". In: *Atmospheric Chemistry and Physics* 10.23, pp. 11373–11383. ISSN: 1680-7324. DOI: 10.5194/acp-10-11373-2010.

Lohmann, Ulrike (2002). "Possible Aerosol Effects on Ice Clouds via Contact Nucleation". In: *JOURNAL OF THE ATMOSPHERIC SCIENCES* 59, p. 10.

Lohmann, Ulrike and Erich Roeckner (1996). "Design and Performance of a New Cloud Microphysics Scheme Developed for the ECHAM General Circulation Model". In: *Climate Dynamics* 12, p. 16.

Lohmann, Ulrike, Johann Feichter, Catherine C. Chuang, and Joyce E. Penner (1999). "Prediction of the Number of Cloud Droplets in the ECHAM GCM". In: *Journal of Geophysical Research: Atmospheres* 104.D8, pp. 9169–9198. ISSN: 01480227. DOI: 10.1029/1999JD900046.

Morales, Annareli, Derek J. Posselt, and Hugh Morrison (2021). "Which Combinations of Environmental Conditions and Microphysical Parameter Values Produce a Given Orographic Precipitation Distribution?" In: *Journal of the Atmospheric Sciences* 78.2, pp. 619–638. ISSN: 0022-4928, 1520-0469. DOI: 10.1175/JAS-D-20-0142.1.

Morris, Max D and Toby J Mitchell (1995). "Exploratory Designs for Computational Experiments". In: *Journal of Statistical Planning and Inference* 43, p. 22.

Morrison, Hugh and Jason A. Milbrandt (2015). "Parameterization of Cloud Microphysics Based on the Prediction of Bulk Ice Particle Properties. Part I: Scheme Description and Idealized Tests". In: *Journal of the Atmospheric Sciences* 72.1, pp. 287–311. ISSN: 0022-4928, 1520-0469. DOI: `10.1175/JAS-D-14-0065.1`.

Morrison, Hugh et al. (2020). "Confronting the Challenge of Modeling Cloud and Precipitation Microphysics". In: *Journal of Advances in Modeling Earth Systems* 12.8. ISSN: 1942-2466, 1942-2466. DOI: `10.1029/2019MS001689`.

Muench, Steffen and Ulrike Lohmann (2020). "Developing a Cloud Scheme With Prognostic Cloud Fraction and Two Moment Microphysics for ECHAM-HAM". In: *Journal of Advances in Modeling Earth Systems* 12.8. ISSN: 1942-2466, 1942-2466. DOI: `10.1029/2019MS001824`.

Pincus, Robert and Stephen A. Klein (2000). "Unresolved Spatial Variability and Microphysical Process Rates in Large-Scale Models". In: *Journal of Geophysical Research: Atmospheres* 105.D22, pp. 27059–27065. ISSN: 01480227. DOI: `10.1029/2000JD900504`.

Posselt, Derek J. (2016). "A Bayesian Examination of Deep Convective Squall-Line Sensitivity to Changes in Cloud Microphysical Parameters". In: *Journal of the Atmospheric Sciences* 73.2, pp. 637–665. ISSN: 0022-4928, 1520-0469. DOI: `10.1175/JAS-D-15-0159.1`.

Reddington, C. L. et al. (2017). "The Global Aerosol Synthesis and Science Project (GASSP): Measurements and Modeling to Reduce Uncertainty". In: *Bulletin of the American Meteorological Society* 98.9, pp. 1857–1877. ISSN: 0003-0007, 1520-0477. DOI: `10.1175/BAMS-D-15-00317.1`.

Regayre, Leighton A. et al. (2018). "Aerosol and Physical Atmosphere Model Parameters Are Both Important Sources of Uncertainty in Aerosol ERF". In: *Atmospheric Chemistry and Physics* 18.13, pp. 9975–10006. ISSN: 1680-7324. DOI: `10.5194/acp-18-9975-2018`.

Rougier, Jonathan, David M. H. Sexton, James M. Murphy, and David Stainforth (2009). "Analyzing the Climate Sensitivity of the HadSM3 Climate Model Using Ensembles from Different but Related Experiments". In: *Journal of Climate* 22.13, pp. 3540–3557. ISSN: 1520-0442, 0894-8755. DOI: `10.1175/2008JCLI2533.1`.

Sengupta, Kamalika, Kirsty Pringle, Jill S. Johnson, Carly Reddington, Jo Browse, Catherine E. Scott, and Ken Carslaw (2021). "A Global Model Perturbed Parameter Ensemble Study of Secondary Organic Aerosol Formation". In: *Atmospheric Chemistry and Physics* 21.4, pp. 2693–2723. ISSN: 1680-7324. DOI: `10.5194/acp-21-2693-2021`.

tisimst (2021). *PyDOE: The Experimental Design Package for Python*.

Tully, Colin, David Neubauer, Nadja Omanovic, and Ulrike Lohmann (2021). *Cirrus Cloud Thinning Using a More Physically-Based Ice Microphysics Scheme in the ECHAM-HAM GCM*. Preprint. Clouds and Precipitation/Atmospheric Modelling/Troposphere/Physics (physical properties and processes). DOI: `10.5194/acp-2021-685`.

van Lier-Walqui, Marcus, Tomislava Vukicevic, and Derek J. Posselt (2014). "Linearization of Microphysical Parameterization Uncertainty Using Multiplicative Process Perturbation Parameters". In: *Monthly Weather Review* 142.1, pp. 401–413. ISSN: 0027-0644, 1520-0493. DOI: `10.1175/MWR-D-13-00076.1`.

van Lier-Walqui, Marcus, Hugh Morrison, Matthew R. Kumjian, Karly J. Reimel, Olivier P. Prat, Spencer Lunderman, and Matthias Morzfeld (2019). "A Bayesian Approach for Statistical–Physical Bulk Parameterization of Rain Microphysics. Part II: Idealized Markov Chain Monte Carlo Experiments". In: *Journal of the Atmospheric Sciences* 77.3, pp. 1043–1064. ISSN: 0022-4928, 1520-0469. DOI: `10.1175/JAS-D-19-0071.1`.

Watson-Parris, Duncan (2021). *GCEm*.

Watson-Parris, Duncan, Andrew Williams, and Pietro Monticone (2021a). *Duncanwp/ESEm: V1.1.0*. Zenodo. DOI: `10.5281/ZENODO.5196631`.

Watson-Parris, Duncan, Andrew Williams, Lucia Deaconu, and Philip Stier (2021b). "Model Calibration Using ESEm v1.0.0 – an Open, Scalable Earth System Emulator". In: *Geoscientific Model Development Discussions*, p. 24. DOI: `10.5194/gmd-2021-267`.

Wellmann, Constanze, Andrew I. Barrett, Jill S. Johnson, Michael Kunz, Bernhard Vogel, Ken S. Carslaw, and Corinna Hoose (2020). "Comparing the Impact of Environmental Conditions and Microphysics on the Forecast Uncertainty of Deep Convective Clouds and Hail". In: *Atmospheric Chemistry and Physics* 20.4, pp. 2201–2219. ISSN: 1680-7324. DOI: `10.5194/acp-20-2201-2020`.

White, Bethan, Edward Gryspeerdt, Philip Stier, Hugh Morrison, Gregory Thompson, and Zak Kipling (2017). "Uncertainty from the Choice of Microphysics Scheme in Convection-Permitting Models Significantly Exceeds Aerosol Effects". In: *Atmospheric Chemistry and Physics* 17.19, pp. 12145–12175. ISSN: 1680-7324. DOI: `10.5194/acp-17-12145-2017`.

Williamson, Daniel, Adam T. Blaker, Charlotte Hampton, and James Salter (2015). "Identifying and Removing Structural Biases in Climate Models with History Matching". In: *Climate Dynamics* 45.5-6, pp. 1299–1324. ISSN: 0930-7575, 1432-0894. DOI: `10.1007/s00382-014-2378-z`.

Yan, Huiping, Yun Qian, Chun Zhao, Hailong Wang, Minghuai Wang, Ben Yang, Xiaohong Liu, and Qiang Fu (2015). "A New Approach to Modeling Aerosol Effects on East Asian Climate: Parametric Uncertainties Associated with Emissions, Cloud Microphysics, and Their Interactions". In: *Journal of Geophysical Research: Atmospheres* 120.17, pp. 8905–8924. ISSN: 2169-897X, 2169-8996. DOI: `10.1002/2015JD023442`.

This document was generated with a layout template provided by Martin Schrön (`github.com/mschroen/review_response_letter`).

---

## Referee Report (RR1)

Second review of "**Assessing the potential for simplification in global climate model microphysics**", by Proske et al., submitted to *ACP*.

**General comments.** It's clear that the authors put considerable effort into revising their paper and it is much improved. Most of my previous comments were addressed. I do have a few broader comments followed by several minor comments/suggestions and technical edits. None of these are major, and my overall recommendation is to accept pending these *minor revisions*. Note all line numbers referred to in this review correspond to the **track-changed version**.

**Semi-major comments.**

1. Overall I appreciate the authors main argument about implications for simplification of process formulations based on the process rate sensitivity analysis/PPE proposed here. However, an essential part in deciding whether or not a process parameterization is a candidate for simplification is the uncertainty in that process. Here the authors perturbed parameters by multiplicative factors from 0. 5 to 2 with equal probability across this region, which doesn't correspond to actual process uncertainty (though to be clear, the authors never claim that it does). The gray area in Fig. 2 is the region "of most interest" over which process sensitivities are considered as candidates for process simplification. However, I feel the authors need to emphasize that this region "of most interest" should ultimately depend on the range of process uncertainty; moreover, this uncertainty range varies from process to process (since, of course, some processes are relatively much more uncertain than others). Thus, sensitivity to process perturbations, as the authors analyze here, should go hand-in-hand with estimating the process uncertainty when considering which processes to simplify. Of course, quantifying process-level uncertainty is itself a major challenge and generally not very straightforward.

2. While I agree about the main arguments concerning simplification, I don't follow how process simplification would lead to greater process understanding as stated on line 73. One could argue that simplification could lead to a greater understanding of the *effects* of a process in a model, but not the process itself. Thus, I suggest rewording this discussion around line 73.

3. The importance of a process as gauged by the method in this paper is also conditional on other parameters/processes that were not varied, including those in parameterizations besides microphysics. I'm sure the authors know this and would agree, but I suggest this be mentioned in the paper. Perhaps around line 574, where they discuss a similar situation with the process analysis being conditional on model resolution.

4. I have some additional questions about interpretation of process simplification depending on the response to the process perturbations. For example, on lines 225-230, the discussion mentioning a critical point with a slope of 0 at eta = 1 seems questionable, since to me this implies a local max/min or perhaps even a cusp. I think instead what you mean is that the slope *around* eta = 1 is small, not that the slope *at* eta = 1 is 0. This is also related to the interpretation given in the Figure 2. I think this figure is useful to include, but if we care about the region in gray, the green and purple lines practically overlay within this region. Thus, it

seems questionable to argue that behavior of the purple line means this process is necessary but doesn't need to be accurate while for the green line the process is dispensable – they practically overlay in the region "of most interest".

Overall, it seems that the key is the average slope of the line within the gray region of interest (the average being consistent with process uncertainty represented as a prior with constant probability across this region). In my view, a near 0 slope across this gray region would imply the process is dispensable, weak slope would be the process is necessary but with less need for accuracy, and moderate to steep slope would mean the process needs accurate representation. Do you agree? The fact that the purple line has a steep slope but it's outside the gray region of interest would seem to imply the steep slope is not relevant. This comment is related to comment #1 above about how the gray region "of most interest" should be tied back to the uncertainty in a specific process (again, varying for different processes).

5. Minor comment but relevant to many places in the text →I'd replace "inflicted" with "induced" throughout the text. "inflicted" usually implies some kind of malevolence (imposing something unwelcome).

**Minor comments.**

Line 32. Similar to a comment in my first review on the Archer-Nicholls et al. (2021) paper specifically focusing on land surface modeling, citing Morrison et al. (2020) seems questionable here since that paper specifically focuses on microphysics and this sentence is a very general statement about the Earth system. I'd either drop the citation here (a citation is not really needed here anyway, this statement is general knowledge), or add "cloud microphysical" before processes if you want this sentence to refer specifically to microphysics.

Line 89. I feel a bit uncomfortable with how this is worded because these papers used synthetic observations (generated by a model) for the constraint. This is stated on line 91 for the Morales et al. (2021) paper but it's not clear to readers this is the case with the other papers as well. Perhaps add "synthetic or real" before "observations" on line 89. Also, you might replace "artificial" with "synthetic" on line 91 for consistency (I think "synthetic" is a slightly better word choice here).

Lines 205-206. With the changes to the text this description now makes much more sense and I actually don't think Eq. (1) is needed. (unfortunately Eq. 1 also introduces a few new issues, like it only makes sense if level number decreases with height but that isn't explicitly stated anywhere). My suggestion is just to remove Eq. 1. now. Also, you refer to level $i$ in this equation but later to process $i$ on p. 9, which I suppose could be confusing. Maybe use $k$ instead of $i$ here?

Line 216. But processes don't affect "tracer variables" (tracer by the usual definition means not impacted by sources/sinks). Would "prognostic variables" be better?

Line 221. Again, "phasing" is confusing terminology here. I'd suggest to reword this "… induced by a process, we can perturb the process using a newly defined parameter eta."

Line 231. I think the wording here is awkward. Suggest instead "…serving as a proxy for understanding sensitivity to processes, while in actuality process uncertainties would likely be variable in time and space."

Line 239. I think you mean the range is expanded to eta = 2 in this second step? Can that be clarified in this sentence?

Line 247. Could replace "thus generated" with "LHS generated", which seems better wording.

Line 437. "slight simplifications" is rather vague. I'd just suggest dropping "slight".

Lines 475-476. For the sentence "Most variables had to be excluded…" could you provide a bit more information or context about this? For example, which variables were kept? What's meant more specifically by "most"?

Line 513. Similar to a comment above, I think "prognostic variables" would be better as "tracers" often has a somewhat different meaning in the literature.

Line 517. Perhaps this is nitpicky, but it's not clear to me how process simplification leads to improved robustness, while the arguments for improved scheme compactness and interpretability are much clearer. Perhaps just deleted "robustness"?

Line 548. Again, "tracer" here is inconsistent with the typical use of the term in atmospheric or climate modeling. Suggest simply removing "tracer" in this sentence.

Lines 552-554. I disagree somewhat with the argument here. Yes, autoconversion generally is associated with the aggregation process (though not always, e.g. Harrington et al. 1995, JAS). However, so are self-collection and accretion, so one has to make some ad-hoc choice, such as a size threshold, to discriminate particle aggregation that leads to self-collection, accretion, or autoconversion. Thus, I agree that autoconversion is "difficult to constrain in observations", but would add that a fundamental challenge in constraining autoconversion is that it is not even a distinct physical process.

Line 554. I don't agree with the argument that this challenge points towards moving to a single ice category scheme, but rather suggests moving to schemes that do not use pre-defined ice categories corresponding to e.g., cloud ice, snow, graupel, whether one category or multiple categories. For instance, there is a multi-category version of the P3 scheme (Milbrandt and Morrison 2016) in which different categories have distinct physical properties, but these properties evolve in time and space and any category could evolve to any ice type depending on local conditions and growth history that the category experiences. Other multi-category

"particle property" based schemes have also been developed that similarly allow multiple categories to evolve to any type of ice (e.g., ISHMAEL, Jensen et al. 2017).

Line 577. I would suggest replacing "in reality" with "other schemes" since processes like autoconversion don't correspond with a distinct process in nature (as argued above, self-collection, accretion, and autoconversion are all associated with the physical process of ice particle aggregation and must be separated using some ad-hoc method like a size threshold). Of course, even the same scheme but in another model, this would likely produce different results with regard to process sensitivity as well.

**Editorial comments.**

Line 58. Could remove "have".

Line 301. I don't follow this sentence, maybe a grammar problem. Should "constrain" be "constraint"?

Line 309. I think there should be "a" before "few".

Lines 350-351. I'd remove "what we call".

Line 497. Suggest changing "less strong or consistent" to "weaker or less consistent".

Line 505. Suggest replacing "can be" with "may be".

Line 567. There is an extra right parenthesis.

**References.**

Harrington, J. Y., M. P. Meyers, R. L. Walko, and W. R. Cotton, 1995: Parameterization of ice crystal conversion process due to vapor deposition for mesoscale models using double-moment basis functions. Part I: Basic formulation and parcel model results. *J. Atmos. Sci.*, 52, 4344–4366.

Jensen, A., J. Harrington, H. Morrison, and J. Milbrandt, 2017: Predicting ice shape evolution in a bulk microphysics model. *J. Atmos. Sci.*, 74, 2081-2104.

Milbrandt, J. A., and H. Morrison, 2016: Parameterization of cloud microphysics based on the prediction of ice particle properties. Part 3: Introduction of multiple free categories. *J. Atmos. Sci.*, 73, 975-995.

---

## Author Response (AR2)

**Author Response to Reviews of**

**Assessing the potential for simplification in global climate model cloud microphysics**

Ulrike Proske, Sylvaine Ferrachat, David Neubauer, Martin Staab, and Ulrike Lohmann Atmospheric Chemistry and Physics, doi:10.5194/acp-2021-801

**RC:** *Reviewer Comment*, AR: *Author Response*,  $\Box$  Manuscript text

We sincerely thank the reviewers for their constructive and detailed feedback. We implemented their feedback into a newly revised version of the manuscript. Please find our answer to the reviewers' points below, followed by a marked-up manuscript version.

**1. Reviewer Comment #1**

- RC: It's clear that the authors put considerable effort into revising their paper and it is much improved. Most of my previous comments were addressed. I do have a few broader comments followed by several minor comments/suggestions and technical edits. None of these are major, and my overall recommendation is to accept pending these minor revisions. Note all line numbers referred to in this review correspond to the track-changed version.
- AR: Thank you for your thorough feedback. Please find our respective answers diretly below your comments.

**1.1. Semi-major comments**

1.1.1

- RC: Overall I appreciate the authors main argument about implications for simplification of process formulations based on the process rate sensitivity analysis/PPE proposed here. However, an essential part in deciding whether or not a process parameterization is a candidate for simplification is the uncertainty in that process. Here the authors perturbed parameters by multiplicative factors from 0. 5 to 2 with equal probability across this region, which doesn't correspond to actual process uncertainty (though to be clear, the authors never claim that it does). The gray area in Fig. 2 is the region "of most interest" over which process sensitivities are considered as candidates for process simplification. However, I feel the authors need to emphasize that this region "of most interest" should ultimately depend on the range of process uncertainty; moreover, this uncertainty range varies from process to process (since, of course, some processes are relatively much more uncertain than others). Thus, sensitivity to process perturbations, as the authors analyze here, should go hand-in-hand with estimating the process uncertainty when considering which processes to simplify. Of course, quantifying process-level uncertainty is itself a major challenge and generally not very straightforward.
- AR: Thank you for your suggestion, which we have gladly incorporated.

If the process uncertainty were known, it would influence the extent of the perturbation range, which could be different for each process.

However, in deciding how drastic these simplifications should be, process uncertainty should also be considered.

**1.1.2**

- RC: While I agree about the main arguments concerning simplification, I don't follow how process simplification would lead to greater process understanding as stated on line 73. One could argue that simplification could lead to a greater understanding of the effects of a process in a model, but not the process itself. Thus, I suggest rewording this discussion around line 73.
- AR: We agree that the gained understanding first and foremost concerns the model itself. We have reworded the sentence accordingly. For the sake of explaining our intentions, what we had meant was that if processes as represented in the model have anything to do with reality, than in turn learning about the model and the processes within might allow to learn more about the processes themselves.

In this paper, we propose a new methodology to assess where process parameterisations can be stripped of detail to aid the development of a simplified model as well as to increase process understanding understanding of the model.

**1.1.3**

- RC: The importance of a process as gauged by the method in this paper is also conditional on other parameters/processes that were not varied, including those in parameterizations besides microphysics. I'm sure the authors know this and would agree, but I suggest this be mentioned in the paper. Perhaps around line 574, where they discuss a similar situation with the process analysis being conditional on model resolution.
- AR: We agree and have added a clarifying remark:

We emphasize that our findings are conditional on the design of the ECHAM-HAM model, including the implementation of other processes and parameters that were not varied in the current study.

**1.1.4**

RC: I have some additional questions about interpretation of process simplification depending on the response to the process perturbations. For example, on lines 225-230, the discussion mentioning a critical point with a slope of 0 at eta = 1 seems questionable, since to me this implies a local max/min or perhaps even

**a cusp. I think instead what you mean is that the slope around eta = 1 is small, not that the slope at eta = 1 is 0.**

AR: Your reasoning is correct: the slope does not need to be 0 exactly and we have modified the statement accordingly:

For example, if the model output variable (e.g. ice water path, IWP) as a function of  $\eta_i$  has a critical point slope close to zero at  $\eta_i = 1$  (i. e. slope of zerogreen and purple lines in Fig. 2), this suggests that the process *i* needs to be represented only approximately and that some detail could probably be removed from its parameterisation without much of an effect on the model performance.

- RC: This is also related to the interpretation given in the Figure 2. I think this figure is useful to include, but if we care about the region in gray, the green and purple lines practically overlay within this region. Thus, it seems questionable to argue that behavior of the purple line means this process is necessary but doesn't need to be accurate while for the green line the process is dispensable they practically overlay in the region "of most interest". Overall, it seems that the key is the average slope of the line within the gray region of interest (the average being consistent with process uncertainty represented as a prior with constant probability across this region). In my view, a near 0 slope across this gray region would imply the process is dispensable, weak slope would be the process is necessary but with less need for accuracy, and moderate to steep slope would mean the process needs accurate representation. Do you agree? The fact that the purple line has a steep slope but it's outside the gray region of interest would seem to imply the steep slope is not relevant. This comment is related to comment #1 above about how the gray region "of most interest" should be tied back to the uncertainty in a specific process (again, varying for different processes).
- AR: The green and purple lines purposefully overlay in the grey region. As you say, it is key that their slope is small here, suggesting that both corresponding processes may be simplified. And as you state correctly, the larger the slope the less simplification is advisable. However, the point we wish to make with the difference between the purple and green line is the following: their small slope in the grey area suggests that they can both be simplified. But because the green line is close to zero throughout the whole phase space, it may even be removed. Thus we chose to give them a similar slope to make that point more clearly. We have added text to the caption to explain our interpretation of the Figure.

Sketch of the envisioned interpretation. The shading indicates the area that is of most interest to judge the effect of process simplifications on the model output. If the slope in this area is small, this suggests that the process can be simplified (green and purple lines). A large slope indicates that the process needs to be represented accurately (orange lines). If no perturbations of the process in the 0.5 to 2 perturbation parameter range and the suppression of the process (perturbatin parameter of 0, not shown) have a significant influence on the model output, the process may be removed entirely (green line).

**1.1.5**

**RC:** Minor comment but relevant to many places in the text: I'd replace "inflicted" with "induced" throughout the text. "inflicted" usually implies some kind of malevolence (imposing something unwelcome).

AR: Thank you for making us aware of these implications. We have substituted all appearances of the word.

**1.2.** Minor comments**

1.2.1 Line 32

- RC: Similar to a comment in my first review on the Archer-Nicholls et al. (2021) paper specifically focusing on land surface modeling, citing Morrison et al. (2020) seems questionable here since that paper specifically focuses on microphysics and this sentence is a very general statement about the Earth system. I'd either drop the citation here (a citation is not really needed here anyway, this statement is general knowledge), or add "cloud microphysical" before processes if you want this sentence to refer specifically to microphysics.
- AR: As suggested we have dropped the citation.

**1.2.2 Line 89**

- RC: I feel a bit uncomfortable with how this is worded because these papers used synthetic observations (generated by a model) for the constraint. This is stated on line 91 for the Morales et al. (2021) paper but it's not clear to readers this is the case with the other papers as well. Perhaps add "synthetic or real" before "observations" on line 89. Also, you might replace "artificial" with "synthetic" on line 91 for consistency (I think "synthetic" is a slightly better word choice here).
- AR: We have added a clarifying note and thank you for your suggestion of the word "synthetic", which indeed is a better choice here.

In a next step, parameter ranges can be constrained when comparing the PPE to observations (Posselt, 2016; van Lier-Walqui et al., 2014; van Lier-Walqui et al., 2019)(Posselt, 2016; van Lier-Walqui et al., 2014; van Lier-Morales et al. (2021) built a PPE of CMP process parameters and environmental conditions, generated using a Markov Chain Monte Carlo algorithm, in idealized simulations to then constrain the parameters with artificial synthetic observations.

**1.2.3 Lines 205-206**

- RC: With the changes to the text this description now makes much more sense and I actually don't think Eq. (1) is needed. (unfortunately Eq. 1 also introduces a few new issues, like it only makes sense if level number decreases with height but that isn't explicitly stated anywhere). My suggestion is just to remove Eq. 1. now. Also, you refer to level i in this equation but later to process i on p. 9, which I suppose could be confusing. Maybe use k instead of i here?
- AR: Since we had added the equation following your suggestion and you now think that the updated description is clear enough, we have removed the equation again.

**1.2.4 Lines 216**

- RC: But processes don't affect "tracer variables" (tracer by the usual definition means not impacted by sources/sinks). Would "prognostic variables" be better?
- AR: Thank you for this suggestion, which we have incorporated.

In the present study, we achieve this by setting to zero the change that the process inflicts on tracer induces on prognostic variables.

**1.2.5 Line 221**

- RC: Again, "phasing" is confusing terminology here. I'd suggest to reword this "... induced by a process, we can perturb the process using a newly defined parameter eta."
- AR: We have incorporated this as you suggested.

More generally, instead of setting to zero the changes inflicted induced by a process, we can phase these changes in and out perturb the process using a newly defined parameter  $\eta$ .

**1.2.6 Line 231**

- **RC:** I think the wording here is awkward. Suggest instead "...serving as a proxy for understanding sensitivity to processes, while in actuality process uncertainties would likely be variable in time and space."
- AR: We have incorporated some of your rewording suggestions but refrain from using the complete wording you suggest as it changes the meaning of the sentence. Here we are discussing that simplifications of processes will likely have an effect that varies in space and time while you are referring to process uncertainties.

Note that the perturbations are constant in space and time for each PPE member, serving as a reasonable proxy for proxy for understanding the effect of possible simplifications, which would likely be variable in space and time time and space.

**1.2.7 Line 239**

- **RC:** I think you mean the range is expanded to eta = 2 in this second step? Can that be clarified in this sentence?
- AR: *This is correct and we have added this specification as follows:*

In addition, the range of  $\eta_i$  is expanded to values up to  $\eta_i = 2$  to imitate an overestimation of a given process due to an inaccurate description.

**1.2.8 Line 247**

- RC: Could replace "thus generated" with "LHS generated", which seems better wording.
- AR: We agree that this is better wording and have replaced as you suggest.

Each of the thus LHS generated input combinations was then used as input for a 1 year ECHAM-HAM model simulation, creating a perturbed parameter ensemble (PPE) with 48 members.

**1.2.9 Line 437**

**RC: "slight simplifications" is rather vague. I'd just suggest dropping "slight".**

AR: We have dropped it as suggested.

Due to the small deviations in the considered variables in response to variations around  $\eta_i = 1$  for riming and autoconversion (purple line in Fig. 2), there is potential for slight simplifications of their formulations.

**1.2.10 Line 475-476**

- RC: For the sentence "Most variables had to be excluded..." could you provide a bit more information or context about this? For example, which variables were kept? What's meant more specifically by "most"?
- AR: We have specified that only 7 variables could be retained (note that this number changed compared to the last version of the manuscript because the spherical harmonics analysis is now applied to absolute values, in response to reviewer comment 2.1.7).

Most 4 out of 11 variables had to be excluded because too many members were defaulting or because their variations were too small to be sensibly emulated.

**1.2.11 Line 513**

- RC: Similar to a comment above, I think "prognostic variables" would be better as "tracers" often has a somewhat different meaning in the literature.
- AR: We have incorporated your suggestion.

Of course, if numerous CMP processes and interactions with aerosols were simplified, this would allow for more drastic steps such as fewer prognostic aerosol tracers variables as those could become redundant.

**1.2.12 Line 517**

- RC: Perhaps this is nitpicky, but it's not clear to me how process simplification leads to improved robustness, while the arguments for improved scheme compactness and interpretability are much clearer. Perhaps just deleted "robustness"?
- AR: Following your suggestion we have deleted it.

However, as detailed in Sec. 1, there are numerous benefits in simplification that are independent of the associated computing cost, such as a gain in compactness , robustness and interpretability.

**1.2.13 Line 548**

- RC: Again, "tracer" here is inconsistent with the typical use of the term in atmospheric or climate modeling. Suggest simply removing "tracer" in this sentence.
- AR: Removing "tracer" lends itself to a broader discussion of the CMP scheme and so we welcome this suggestion.

For example, CMP schemes that contain only one tracer category for ice, e.g. the Predicted Particle Properties (P3) ice microphysics scheme (e.g. Morrison and Milbrandt (2015), Eidhammer et al. (2017), Dietlicher et al. (2018), Dietlicher et al. (2019), and Tully et al. (2021)) are more physical as well as more interpretable.

**1.2.14 Line 552-554**

- RC: I disagree somewhat with the argument here. Yes, autoconversion generally is associated with the aggregation process (though not always, e.g. Harrington et al. 1995, JAS). However, so are self-collection and accretion, so one has to make some ad-hoc choice, such as a size threshold, to discriminate particle aggregation that leads to self-collection, accretion, or autoconversion. Thus, I agree that autoconversion is "difficult to constrain in observations", but would add that a fundamental challenge in constraining autoconversion is that it is not even a distinct physical process.
- AR: In the ECHAM-HAM CMP scheme, autoconversion is associated only with the aggregation process (as depicted in Fig. 1). We think we have covered the point of the artificial size threshold in this discussion by stating that autoconversion "is a transfer mechanism between the two artificial classes". Still, we thank you for your suggestion and have now added a remark stressing that autoconversion is not a distinct physical process.

Still, autoconversion is difficult to constrain in observations (Morrison et al., 2020) also because it is not a distinct physical process, and so moving towards a one ice category scheme seems advisable. scheme with evolving instead of pre-defined ice categories seems advisable (see e.g. Milbrandt and Morrison (2016) and Jensen et al. (2017)).

**1.2.15 Line 554**

- RC: I don't agree with the argument that this challenge points towards moving to a single ice category scheme, but rather suggests moving to schemes that do not use pre-defined ice categories corresponding to e.g., cloud ice, snow, graupel, whether one category or multiple categories. For instance, there is a multicategory version of the P3 scheme (Milbrandt and Morrison 2016) in which different categories have distinct physical properties, but these properties evolve in time and space and any category could evolve to any ice type depending on local conditions and growth history that the category experiences. Other multi-category" particle property" based schemes have also been developed that similarly allow multiple categories to evolve to any type of ice (e.g., ISHMAEL, Jensen et al. 2017).
- AR: We completely agree with your point and have gladly added it and the references you point to.

Still, autoconversion is difficult to constrain in observations (Morrison et al., 2020) also because it is not a distinct physical process, and so moving towards a one-ice category scheme seems advisable. scheme with evolving instead of pre-defined ice categories seems advisable (see e.g. Milbrandt and Morrison (2016) and Jensen et al. (2017)).

**1.2.16 Line 577**

- RC: I would suggest replacing "in reality" with "other schemes" since processes like autoconversion don't correspond with a distinct process in nature (as argued above, self- collection, accretion, and autoconversion are all associated with the physical process of ice particle aggregation and must be separated using some ad-hoc method like a size threshold). Of course, even the same scheme but in another model, this would likely produce different results with regard to process sensitivity as well.
- AR: We agree that the results cannot be transferred to other schemes and that in reality the processes investigated here do not exist as such. However, we would like to insist that this is a point worth making and have therefore left the remark about reality as well as adding the other schemes.

Thus the results as such are only applicable to this CMP scheme and cannot be transferred to the significance of the investigated processes in other schemes let alone in reality.

**1.3.** Editorial comments**

- Line 58. Could remove "have".
- Line 301. I don't follow this sentence, maybe a grammar problem. Should "constrain" be "constraint"?
- Line 309. I think there should be "a" before "few".
- Lines 350-351. I'd remove "what we call".
- Line 497. Suggest changing "less strong or consistent" to "weaker or less consistent".
- Line 505. Suggest replacing "can be" with "may be".
- Line 567. There is an extra right parenthesis.

AR: Thank you for these editorial remarks. We have incorporated your points as suggested.

**2. Reviewer Comment #2**

- RC: I would like to thank the authors for addressing the key points from my first review of this paper to a high standard. The PPE design now covers the parameter space considered much more evenly, leading to a more robust emulator and hence resulting analysis. This is a novel use of the emulation and sensitivity analysis approach to assess model behaviour under uncertainty. I have a few further minor comments, but once these are addressed I would recommend the publication of the manuscript in ACP.
- AR: Thank you for your again very thoughtful feedback. Please find our answers to your comments below.

**2.1. Specific Comments**

2.1.1 Line 139-144

- **RC:** Here, the sections of the paper are introduced, but there is no mention of the sections relating to the seasonal and spatially resolved analysis, which is mentioned in the abstract? Perhaps add a sentence here to also point to these for easy reference?
- AR: Thank you for pointing out these missing references, which we have now included.

In Sect. 3 the results from a "one-at-a-time" sensitivity study that explores the axes of the parameter space (Sect. 3.1), the emulated PPE (Sect. 3.2), and of the sensitivity study on the fully sampled parameter space (Sect. 3.3) including a scale dependency (Sect. 3.4) and seasonal analysis (Sect. 3.5) are presented and discussed. Conclusions and an outlook are given in Sect. 4.

**2.1.2 Figure 2 and Section 2.2**

- RC: I like the new Figure 2 that has been added to the manuscript to help with the interpretation on how the effects of the perturbations point to potential for process simplification. However, there is no link or reference to the new Figure 2 in the text of this section, or any text description of this interpretation, so it's not clear. And, the caption just says 'Sketch of the envisioned interpretation...', but interpretation of what? Please add some informative description as to what Figure 2 shows in this section and make the caption clearer as to the interpretation it corresponds to.
- AR: Indeed we had forgotten to link to the Figure in the text, which we have now adjusted. Following your suggestion we have also extended the Figure caption.

In our case, the parameters aid to understand the sensitivity of the model to each process: From the response of model output to variations in  $\eta_i$ , we can extract information on how accurately a process *i* needs to be represented in the model  $\cdot$  (see Fig. 2 for a visualisation). For example, if the model output variable (e.g. ice water path, IWP) as a function of  $\eta_i$  has a eritical point slope close to zero at  $\eta_i = 1$  (i. e. slope of zerogreen and purple lines in Fig. 2), this suggests that the process *i* needs to be represented only approximately and that some detail could probably be removed from its parameterisation without much of an effect on the model performance.

Sketch of the envisioned interpretation. The shading indicates the area that is of most interest to judge the effect of process simplifications on the model output. If the slope in this area is small, this suggests that the process can be simplified (green and purple lines). A large slope indicates that the process needs to be represented accurately (orange lines). If no perturbations of the process have a significant influence on the model output, the process may be removed entirely (green line).

**2.1.3 Line 297-298 (and elsewhere: L289, L469)**

- RC: '... only 47 PPE members were used as with the 48th member the computational constraint was too tight for the emulator'. This is very ambiguous, and I don't think a general reader will understand what is meant by this and so will find this statement confusing. If you need to state that there were issues with the emulator construction, then it needs to be phrased more directly as that...something like: '... only 47 PPE members were used due to instabilities in the computations when constructing the emulator. 'I am a little surprised that this happens, with <50 data points over a 4-d parameter space. For my interest - Is it a specific run that causes this? i.e. the same run each time, so some sort of outlier in your PPE?
- AR: We have used your suggested phrasing as indeed the problems seem to originate in numerical instabilities of the emulator construction. This is supported by the fact that it is not a specific outlier causing problems. In fact, in cases where the model cannot be trained with 48 members, training with any 47 members works.

The difficulty in emulating the response surface for some of the variables was also apparent in computational limitations: some of the leave-one-out validation emulations were not possible to compute because the constrain of the emulator was too tight for the variability in the data. of numerical instabilities in the computations when constructing the emulator. As these were only a few cases (up to six for global means and four for seasonal means in 48 validation emulations), the validation for those variables as a whole is still deemed valid.

For the variables which passed the leave-one-out validation, the final emulator used for the sensitivity analysis was trained on all PPE members (note that in a few cases only 47 PPE members were used as with the 48th member the computational constraint was too tight for the due to numerical instabilities in the computations when constructing the emulator).

**2.1.4 Line 327 (and throughout Section 3.1, and in Section 3.3, 3.6)**

- RC: '... that it's inhibition leads to ...' I mentioned this in my previous review, and I still don't fully understand the meaning of the word 'inhibition' when describing the parameter effects. What is a parameters' inhibition? Do you mean it has very little effect? Or switching it off? – it's not clear. [When I google the meaning of this word, I don't find a relevant meaning for this context.] If you must use this word, then please define what it means before you first use it. Or alternatively, re-phrase the sentences to be clear in meaning and take it out.
- AR: What we mean by inhibition of a process is switching it off. In search of a better word we have replaced inhibition by suppression throughout the text.

**2.1.5 Figure 7 caption**

- RC: The projections of the sampling here are 2-d, not 3-d. And the response surface is a 4-d response surface, not 5-d. The response variable (here, IWP) is not a dimension of the sampling – You only have 4 input parameters that the sampling is over, and so the dimension of the response surface has to be 4-d, as it is showing how the response variable changes over those 4-dimensions of parameter space. Each individual plot here considers 2 of those input dimensions, and therefore it is showing the response over a 2-d projection on the space. Please amend the caption.
- AR: We have amended the caption as you suggested.

Three Two dimensional projections of the sampling of IWP values sampled from the five four dimensional response surface parameter space of the emulated PPE. Each perturbed process is a dimension, and the colorbar denotes the global annual mean ice water path for each input parameter combination.

**2.1.6 Figure 8 caption**

**RC: Please remove the term 'correlation panel' from this caption – see previous review.**

AR: Please excuse that we had misunderstood your previous comment. We have adapted the Figure caption as suggested.

Same as Fig. 7 but for the global annual mean liquid water path. Correlation panels Results for additional variables are presented in Fig. B1 in Appendix B.

**2.1.7 Line 445-446**

**RC: Why do you need to use the difference to the control simulation for this analysis? I don't understand what that achieves ... Could it affect (reduce) the amount of signal that you see?**

AR: Thank you for bringing up this question. Indeed, there is no need to use the difference to the control simulation in this analysis. To do so made sense for us during the development of the analysis but for consistency we have now changed it so that the absolute values are used.

**2.1.8 Figures 11 and 12**

- RC: Following a comment in my previous review I think it would be more informative to show the first-order effects in these figures and have the total effect figures in the appendix. If the first order effect plots look similar to the total effect plots then they are more informative, as with the total effect, the reader is left to ponder/guess whether some of the effect is in fact interaction, when it's probably not? The first order effects correspond to individual parameter effects alone, and so are surely more informative and conclusive here?
- AR: As you suggested we have exchanged the total and first order effect plots.

**2.1.9 Line 481**

**RC: I'm not sure where the value of 0.2% comes from – please clarify.**

AR: We have added a clarifying remark.

They show that at most, with naively removing (the most drastic simplification) the whole cold precipitation formation routine, only about 0.2% of total computing time can be saved (since the cold precipitation formation routine makes up 4.8% of the 4.7% of computing time that the whole CMP take up, see Table 1).

**2.2. Technical Corrections**

- Line 289: Change '... As these were only few cases ...' to '... As these were only a few cases ...'
- Line 326: I know IAV is defined on page 10, but I got to the acronym here and couldn't remember what it meant ... maybe give the full wording here as a reminder? In fact, do you really need to use an acronym for this, given it only appears 3 times?
- Line 372: Change 'Only for LWP Lohmann and Ferrachat (2010) find...' to 'Only for LWP do Lohmann and Ferrachat (2010) find...'
- Figure 12: The plots in the figure need to be labelled to indicate which is plot a), b), c) and d).
- AR: Thank you for these corrections, which we have implemented as suggested.

**References**

Dietlicher, Remo, David Neubauer, and Ulrike Lohmann (2018). "Prognostic Parameterization of Cloud Ice with a Single Category in the Aerosol-Climate Model ECHAM(v6.3.0)-HAM(v2.3)". In: *Geoscientific Model Development* 11.4, pp. 1557–1576. ISSN: 1991-9603. DOI: 10.5194/gmd-11-1557-2018.
— (2019). "Elucidating Ice Formation Pathways in the Aerosol-Climate Model ECHAM6-HAM2". In: *Atmospheric Chemistry and Physics* 19.14, pp. 9061–9080. ISSN: 1680-7324. DOI: 10.5194/acp-19-9061-2019.

- Eidhammer, Trude, Hugh Morrison, David Mitchell, Andrew Gettelman, and Ehsan Erfani (2017). "Improvements in Global Climate Model Microphysics Using a Consistent Representation of Ice Particle Properties". In: *Journal of Climate* 30.2, pp. 609–629. ISSN: 0894-8755, 1520-0442. DOI: 10.1175/JCLI-D-16-0050.1.
- Jensen, Anders A., Jerry Y. Harrington, Hugh Morrison, and Jason A. Milbrandt (2017). "Predicting Ice Shape Evolution in a Bulk Microphysics Model". In: *Journal of the Atmospheric Sciences* 74.6, pp. 2081–2104. ISSN: 0022-4928, 1520-0469. DOI: 10.1175/JAS-D-16-0350.1.
- Milbrandt, J. A. and H. Morrison (2016). "Parameterization of Cloud Microphysics Based on the Prediction of Bulk Ice Particle Properties. Part III: Introduction of Multiple Free Categories". In: *Journal of the Atmospheric Sciences* 73.3, pp. 975–995. ISSN: 0022-4928, 1520-0469. DOI: 10.1175/JAS-D-15-0204.1.
- Morales, Annareli, Derek J. Posselt, and Hugh Morrison (2021). "Which Combinations of Environmental Conditions and Microphysical Parameter Values Produce a Given Orographic Precipitation Distribution?" In: *Journal of the Atmospheric Sciences* 78.2, pp. 619–638. ISSN: 0022-4928, 1520-0469. DOI: 10.1175/JAS-D-20-0142.1.
- Morrison, Hugh and Jason A. Milbrandt (2015). "Parameterization of Cloud Microphysics Based on the Prediction of Bulk Ice Particle Properties. Part I: Scheme Description and Idealized Tests". In: *Journal of the Atmospheric Sciences* 72.1, pp. 287–311. ISSN: 0022-4928, 1520-0469. DOI: 10.1175/JAS-D-14-0065.1.
- Morrison, Hugh et al. (2020). "Confronting the Challenge of Modeling Cloud and Precipitation Microphysics". In: *Journal of Advances in Modeling Earth Systems* 12.8. ISSN: 1942-2466, 1942-2466. DOI: 10.1029/ 2019MS001689.
- Posselt, Derek J. (2016). "A Bayesian Examination of Deep Convective Squall-Line Sensitivity to Changes in Cloud Microphysical Parameters". In: *Journal of the Atmospheric Sciences* 73.2, pp. 637–665. ISSN: 0022-4928, 1520-0469. DOI: 10.1175/JAS-D-15-0159.1.
- Tully, Colin, David Neubauer, Nadja Omanovic, and Ulrike Lohmann (2021). Cirrus Cloud Thinning Using a More Physically-Based Ice Microphysics Scheme in the ECHAM-HAM GCM. Preprint. Clouds and Precipitation/Atmospheric Modelling/Troposphere/Physics (physical properties and processes). DOI: 10.5194/acp-2021-685.
- van Lier-Walqui, Marcus, Tomislava Vukicevic, and Derek J. Posselt (2014). "Linearization of Microphysical Parameterization Uncertainty Using Multiplicative Process Perturbation Parameters". In: *Monthly Weather Review* 142.1, pp. 401–413. ISSN: 0027-0644, 1520-0493. DOI: 10.1175/MWR-D-13-00076.1.
- van Lier-Walqui, Marcus, Hugh Morrison, Matthew R. Kumjian, Karly J. Reimel, Olivier P. Prat, Spencer Lunderman, and Matthias Morzfeld (2019). "A Bayesian Approach for Statistical–Physical Bulk Parameterization of Rain Microphysics. Part II: Idealized Markov Chain Monte Carlo Experiments". In: *Journal* of the Atmospheric Sciences 77.3, pp. 1043–1064. ISSN: 0022-4928, 1520-0469. DOI: 10.1175/JAS– D-19-0071.1.

This document was generated with a layout template provided by Martin Schrön (github.com/mschroen/review\_response\_letter).